# Estimating multiplicity of infection, haplotype frequencies, and linkage disequilibria from multi-allelic markers for molecular disease surveillance

Henri Christian Junior Tsoungui Obama[1,2]*, Kristan Alexander Schneider[1,3,4]

**1** Department of Applied Computer- and Biosciences, University of Applied Sciences Mittweida, Mittweida, Germany, **2** Department of Mathematics, Chemnitz University of Technology, Chemnitz, Germany, **3** Center for Global Health, Department of Internal Medicine, School of Medicine, The University of New Mexico, Albuquerque, New Mexico, United States of America, **4** Translational Informatics Division, Department of Internal Medicine, School of Medicine, The University of New Mexico, Albuquerque, New Mexico, United States of America

☯ These authors contributed equally to this work.

* hts@hrz.tu-chemnitz.de

**Data availability statement:** Only previously published and simulated data was used in the

## Abstract

**Background:** Molecular/genetic methods are becoming increasingly important for surveillance of diseases like malaria. Such methods allow monitoring routes of disease transmission or the origin and spread of variants associated with drug resistance. A confounding factor in molecular disease surveillance is the presence of multiple distinct variants in the same infection (multiplicity of infection – MOI), which leads to ambiguity when reconstructing which pathogenic variants are present in an infection. Heuristic approaches often ignore ambiguous infections, which leads to biased results.

**Methods:** To avoid bias, we introduce a statistical framework to estimate haplotype frequencies alongside MOI from a pair of multi-allelic molecular markers. Estimates are based on maximum likelihood using the expectation-maximization (EM)-algorithm. The estimates can be used as plug-ins to construct pairwise linkage disequilibrium (LD) maps. The finite-sample properties of the proposed method are studied by systematic numerical simulations. These reveal that the EM-algorithm is a numerically stable method in our case and that the proposed method is accurate (little bias) and precise (small variance) for a reasonable sample size. In fact, the results suggest that the estimator is asymptotically unbiased. Furthermore, the method is appropriate to estimate LD (by $D'$, $r^2$, $Q^*$, or conditional asymmetric LD). Furthermore, as an illustration, we apply the new method to a previously published dataset from Cameroon concerning sulfadoxine-pyrimethamine (SP) resistance. The results are in accordance with the SP drug pressure at the time and the observed spread of resistance in the country, yielding further evidence for the adequacy of the proposed method.

**Conclusion:** The proposed method can be readily applied in practice for malaria disease surveillance as a replacement for heuristic methods. The first benefit is its ability to estimate MOI, which scales with transmission intensities, and, in a temporal context,

study. The simulated data can be regenerated using the available code on GitHub (https://github.com/Maths-against-Malaria/ MultiAllelicBiLociModel) and Zenodo (https://doi.org/10.5281/zenodo.8289710).

**Funding:** This study was supported in the form of funding by the German Academic Exchange (Project-ID 57417782, Projekt-ID 57599539) awarded to KAS, Sächsisches Staatsministerium für Wissenschaft und Kunst (Project numbers 100257255 and 100613388) awarded to KAS, the Federal Ministry of Education and Research (BMBF) and the DLR (Project-ID 01DQ20002) awarded to KAS. The funders played no role in the design of the study. There was no additional external funding received for this study.

**Competing interests:** The authors declare that no competing interests exist.

can be used to evaluate the effectiveness of disease control measures. MOI is best estimated from molecular markers that are not under selection (neutral markers) and exhibit sufficient genetic variation. The second advantage is that it can estimate pairwise LD without deflating sample size as in heuristic methods, thereby limiting uncertainty in the estimates. This is particularly useful when deriving LD maps from data with many ambiguous observations due to MOI. Importantly, the method *per se* is not restricted to malaria, but applicable to any disease with a similar transmission pattern. The method and several extensions are implemented in an easy-to-use R script.

## Introduction

Genomic/molecular techniques allow the detection of pathogens with better granularity than the traditional clinical methods and, therefore, enhance disease surveillance [1,2]. Molecular surveillance of infectious diseases allows for retrieving and reconstructing population or evolutionary genetic information such as the spread of drug-resistant pathogenic variants, or detailed epidemiological information such as routes of transmission and transmission intensities [3]. Such information helps to optimize disease control and prevention and becomes increasingly popular for some of the economically most relevant infectious diseases, such as malaria [4,5].

A number of programs targeting the reduction of the global malaria burden have been launched in the past decade, e.g., the President's Malaria Initiative (PMI), the UN Millennium Development Goals (MDGs) followed by the UN Sustainable Development Goals (SDGs; Goal 3 Good Health and Well-Being) accompanied by the Global Technical Strategy For Malaria [6,7]. Despite substantial reductions in malaria incidence in the past decades, the trend has been reversed since 2018. With an estimated 247 million infections, and, 621 000 deaths worldwide in 2022 the disease remains highly endemic, particularly in sub-Saharan Africa, and parts of Oceania (i.e., Papua New Guinea), and South America [8]. This challenges the goal of malaria elimination in at least 30 countries by 2030.

Challenges in successful malaria control are, e.g., the spread of drug resistance, or migration events, by which the disease is reintroduced into areas aiming for malaria elimination and causes local disease outbreaks [9,10]. Both, the spread of drug resistance and migration events cause characteristic patterns of linkage-disequilibrium (LD) which are informative about the underlying evolutionary-genetic processes [11]. This emphasizes the importance of harnessing LD measures in malaria molecular surveillance.

Pairwise LD maps require reliable haplotype frequency estimates. Such estimates are not straightforward in malaria, because multiple genetically distinct pathogenic variants can be present within the same infection, often referred to as complexity of infection (COI) or multiplicity of infection (MOI) [12–14]. Because of this confounding factor, *Plasmodium* (although a haploid organism) 'behaves' as a polyploid organism with a random level of ploidy within human infections.

The characteristic malaria transmission cycle in combination with MOI leads to a departure from classical population genetics [15,16]. Notably, malaria transmission involves a step of meiotic recombination in the mosquito vector, implying that *Plasmodium* acts as a diploid organism within the mosquito. Note that the terms COI or MOI are defined ambiguously in the literature. Here, we follow the stringent definition of [17] of MOI as the number of super-infections, i.e., the number of infective events with the same or different pathogenic variants during one disease episode.

Therefore, care has to be taken when estimating the population genetics of such diseases, as it differs from classical population genetics due to MOI. For instance, MOI plays an important role in the estimation of the frequency of pathogenic variants, and more importantly in the differentiation between pathogens frequencies and prevalence [17].

Typically, statistical models are required to estimate the frequency of each haplotype in the pathogen population from molecular data, while accounting for MOI. Note that, in low transmission areas, only a relatively small amount of infections occurs, yielding molecular dataset with small sample sizes. Moreover, molecular data usually contains missing values as a result of failures in molecular assays and errors in the determination of alleles at markers of concern [17]. The amount of missing values per sample increases with the number of molecular markers considered. Therefore, to minimize the loss of data, fewer loci can be considered. For studying pairwise LD, it suffices to consider two marker loci.

Several statistical methods have been developed to estimate haplotype frequencies alongside MOI in malaria from molecular data. The maximum-likelihood (ML)-based method of [18] estimates haplotype frequencies and MOI, considering one polymorphic marker locus. Another maximum-likelihood method, developed by [19] estimates haplotype frequencies alongside MOI from multiple bi-allelic SNPs and presents a framework to estimate haplotype prevalence using the frequency and MOI estimates as plugin parameters. The method is of particular interest if applied to estimate the frequency of drug-resistant haplotypes in malaria, which are typically characterized by five to ten SNPs [20]. Although this method allows monitoring the spread of drug resistance, mining the patterns of selection caused by drug-resistance evolution (genetic hitchhiking, LD) requires different molecular data and analysis methods. Variable molecular markers such as STRs are useful in this regard, as they are fast evolving and allow to investigate ongoing evolutionary processes such as drug resistance evolution. To study genetic hitchhiking in terms of heterozygosity, it is sufficient to estimate allele-frequency spectra separately for a set of markers flanking the loci under selection, e.g., by the method of [18]. To study pairwise LD maps, haplotypes determined by two multi-allelic markers have to be considered.

For this purpose, we introduce a statistical model that allows to estimate haplotypes frequencies and MOI by ML, assuming two multi-allelic loci. Because no explicit solution for the maximum-likelihood estimate (MLE) can be found, we employ the expectation maximization (EM)-algorithm (cf. [19,21]) to obtain a numerically efficient and stable method to derive the estimates. A systematic simulation study is performed, to evaluate the finite sample properties of the MLE. Furthermore, we investigate the capability of the model to estimate the multi-allelic LD measures $D'$, $Q^*$, and $r^2$ (cf. [22]).

As an example, the method is applied to a molecular dataset from *P. falciparum*-positive ($n = 166$) blood samples collected in Yaoundé, Cameroon from 2001-2002 and 2004-2005 [11]. The patterns of pairwise LD around the dhfr and dhps genes, responsible for sulfadoxine-pyrimethamine (SP) resistance, are studied in terms of mulltiallelic LD measures including the conditional asymmetric LD measure proposed by [23].

An implementation of the maximum-likelihood method is provided as an easy-to-use R script accessible via GitHub https://github.com/Maths-against-Malaria/MultiAllelicBiLociModel and Zenodo at https://doi.org/10.5281/zenodo.8289710.

To enhance readability, technical details are presented in specific sections and the supplementary materials. Fig. 1 gives an overview of the manuscript.

The statistical model is described in Methods, while the model's performance and data applications are described in Results. Readers primarily interested in the applications shall

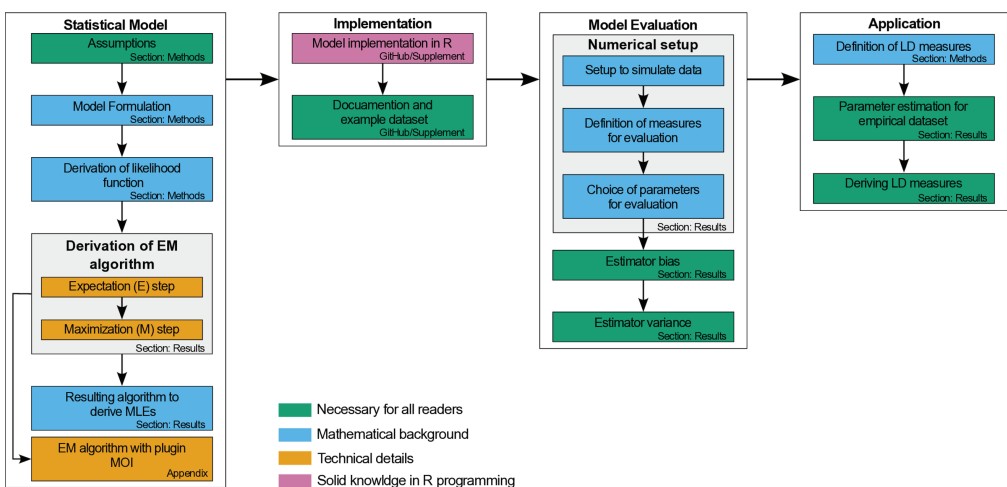

**Fig 1. Overview of the manuscript:** The diagram serves as a map to navigate through the manuscript and indicates, which parts require special background knowledge in mathematics and programming.

go straight to Results. However, readers interested in the mathematical details, find a concise description of all derivations in S1 Mathematical appendix.

## Methods

The statistical model described here adapts the framework of [19] to estimate MOI and haplotype frequencies to the case of two multi-allelic loci, e.g., microsatellite markers. Importantly, we define MOI as the number of super-infections, i.e., independent infectious events that occur during the course of the disease, while assuming no co-transmission (co-infections). The difference between super- and co-infections is illustrated in Fig. 2).

### Statistical model

Let us assume pathogen haplotypes, denoted $\boldsymbol{h}$, characterized by two loci (markers) with $n_1$, $n_2$ alleles, at the first and second locus, respectively. A haplotype is represented by a vector of length two indicating its allelic configuration at each locus, i.e., $\boldsymbol{h} = (h_1, h_2)$, with $h_1 \in \{0, \dots, n_1 - 1\}$ and $h_2 \in \{0, \dots, n_2 - 1\}$. A total of $H = n_1 n_2$ haplotypes are possible with such a genetic architecture. We denote the set of all possible haplotypes by $\mathcal{H}$, hence,

$$\boldsymbol{h} \in \mathcal{H} = \{0, \dots, n_1 - 1\} \times \{0, \dots, n_2 - 1\}.$$

We order the $H$ possible haplotypes by the numbers $1, \dots, H$, such that haplotype $\boldsymbol{h}$ corresponds to the number $[\boldsymbol{h}]_{n_1, n_2} := n_2 h_1 + h_2 + 1$. (As an example, for $n_1 = 2, n_2 = 4$ haplotype $(1,2)$ corresponds to the number: $4 \cdot 1 + 2 + 1 = 4 + 2 + 1 = 7$.)

The frequency of haplotype $\boldsymbol{h}$, denoted by $p_{\boldsymbol{h}}$, is its relative abundance in the pathogen population. Considering all possible haplotypes, we denote haplotype frequencies with the vector $\boldsymbol{p} := (p_{\boldsymbol{h}})_{\boldsymbol{h} \in \mathcal{H}} = (p_1, \dots, p_H)$, where $p_k = p_{\boldsymbol{h}}$ if $[\boldsymbol{h}]_{n_1, n_2} = k$. Note that in practice, among the $H$ possible haplotypes, only a few are present in the pathogen population, i.e., $p_k = 0$ for most haplotypes (see Fig 3A,B).

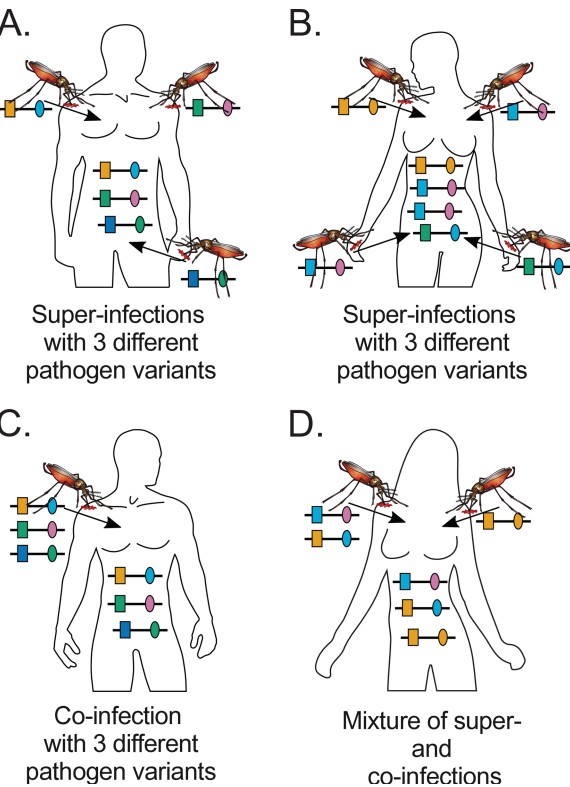

**Fig 2. Super- and co-infections:** The figure illustrates super- and co-infections. **(A)** Illustrated is a super infection by 3 mosquitoes (MOI=3), which transmit 3 different haplotypes. **(B)** Shown is a super-infection by 4 mosquitoes (MOI=4), which transmit 3 distinct haplotypes, as two mosquitoes infect with the same haplotype. **(C)** A co-infection with three different haplotypes transmitted by one mosquito. MOI remains undefined here, as co-infections are ignored. **(D)** A mixture of co-infections ans super-infections. Two mosquitoes transmit haplotypes. One mosquito co-infects with 2 distinct haplotypes, while the super-infecting mosquito infects with one haplotype. Also here MOI remains undefined, as co-infections are not onsidered.

Here, assuming only disease-positive individuals, the number of (super-)infections during the same disease episode, referred to as multiplicity of infections (MOI), follows a conditional (or positive) Poisson distribution [19,24]. The probability that an individual is (super)-infected exactly $m$ times (MOI=$m$) is,

$$\kappa_m = \frac{1}{e^\lambda - 1} \frac{\lambda^m}{m!}, \quad m = 1, 2, 3, \ldots, \tag{1a}$$

where $\lambda$ is the parameter of the distribution. Moreover, the probability generating function (PGF) of the distribution is

$$G(z) := \mathbb{E}[z^m] = \sum_{m=1}^{\infty} \kappa_m z^m = \frac{e^{\lambda z} - 1}{e^\lambda - 1}, \tag{1b}$$

and the mean MOI is (cf. [24])

$$\psi := \mathbb{E}[m] = \frac{\lambda}{1 - e^{-\lambda}}. \tag{1c}$$

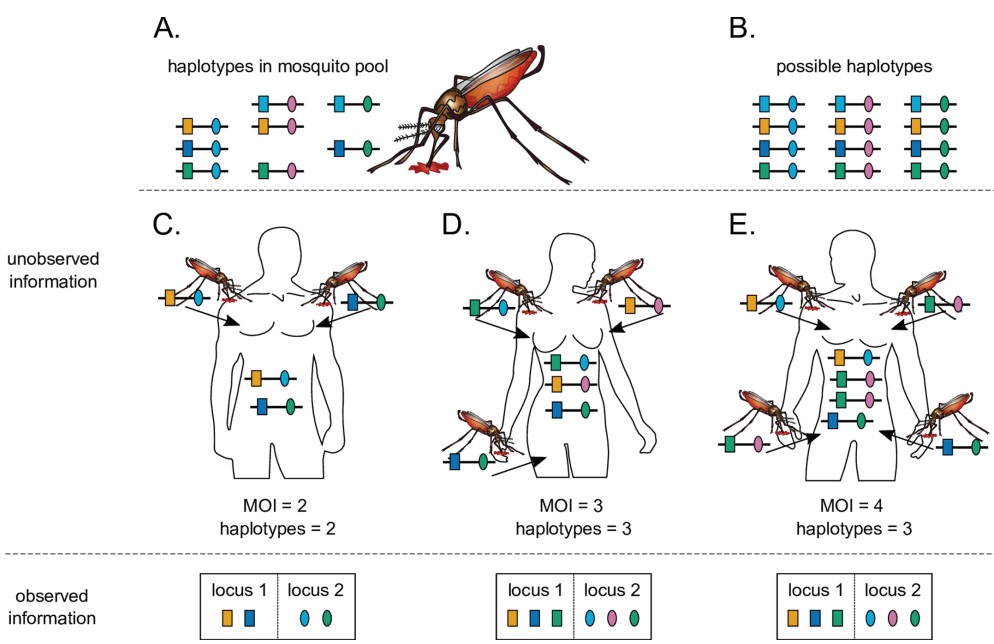

**Fig 3. Super-infections and haplotypes phasing:** Illustrated are different infective events with haplotypes from the same pathogen population. Panel (**A**) shows the haplotypes present in the parasites population, with each haplotype identified by two loci markers (different shapes), and several alleles at each locus (different colors). Considering the number of loci, and the alleles observed at each locus, there is in theory more haplotype that could be present in mosquito population. All the possible haplotypes are shown in panel (**B**). Panel (**C**) illustrates an infection with two different infecting haplotypes (i.e., MOI $m = 2$). The corresponding haplotype information (bottom-left) is ambiguous as it is impossible to reconstruct the infecting haplotypes with certainty. Panel (**D**) is similar to Panel (**C**) with three different infecting haplotypes (MOI $m = 3$). The corresponding haplotype information is ambiguous in this case as well (bottom-middle). (**E**) illustrates a super-infection with three different infecting haplotypes, with one of the haplotypes infecting twice (i.e., MOI $m = 4$). The haplotype information in this case is also ambiguous. Note that, in all cases, it is impossible to be confident in identifying how many times the hosts were infected. MOI information is typically ambiguous.

During an infective event, only one haplotype is randomly sampled from the pathogen population and transmitted to the host [19,24]. Assuming a super-infection with MOI = $m$, an individual is infected $m_{\boldsymbol{h}}$ times with haplotype $\boldsymbol{h}$ (see Fig 3C-E). The infection is described by $\boldsymbol{m} := (m_{\boldsymbol{h}})_{\boldsymbol{h} \in \mathcal{H}} = (m_1, \ldots, m_H)$ such that $|\boldsymbol{m}| := \sum_{\boldsymbol{h} \in \mathcal{H}} m_{\boldsymbol{h}} = m_1 + \ldots + m_H = m$. Therefore, given MOI $m$, the probability of infection $\boldsymbol{m}$ is

$$P(\boldsymbol{m} \mid m) = \binom{m}{\boldsymbol{m}} \boldsymbol{p}^{\boldsymbol{m}}, \tag{2}$$

where $\binom{m}{\boldsymbol{m}} = \frac{m!}{m_1! \ldots m_H!}$, and $\boldsymbol{p}^{\boldsymbol{m}} = p_1^{m_1} \ldots p_H^{m_H}$.

The infection vector $\boldsymbol{m}$ and the corresponding MOI $m$ are unknown in practice since they cannot be observed from a clinical specimen (see Fig 3C-E). Moreover, due to the lack of phasing, the presence of multiple different haplotypes in a clinical sample yields ambiguous genetic information [19,25]. Here, we assume that the absence/presence of alleles at considered loci is the only available information.

Given an infection, its allelic information is denoted by the vector $\boldsymbol{x} = (x_1, x_2)$, where $x_1$ and $x_2$ are the set of alleles detected at the first and second locus, respectively. We assume that all alleles present in an infection are detected and there are no erroneous detection. The allelic

information $x_1$, is a subset of all alleles detectable at the first locus. Therefore, $x_1$ is an element of the powerset of $\{0, \dots, n_1 - 1\}$, i.e., $x_1 \in \mathcal{P}(\{0, \dots, n_1 - 1\}) \setminus \varnothing$. Since only disease-positive samples are considered, the empty set is excluded. Equivalently, $x_2 \in \mathcal{P}(\{0, \dots, n_2 - 1\}) \setminus \varnothing$. Therefore, the set of all possible observations is given by the following cartesian product

$$\mathcal{O} := \left( (\mathcal{P}(\{0, \dots, n_1 - 1\}) \setminus \varnothing) \times (\mathcal{P}(\{0, \dots, n_2 - 1\}) \setminus \varnothing) \right).$$

There is a total of $(2^{n_1} - 1)(2^{n_2} - 1)$ possible observations. Note that a particular observation $\boldsymbol{x}$ can result from different infections $\boldsymbol{m}$, i.e., $\boldsymbol{m} \to \boldsymbol{x}$ (cf. Fig 3D-E). Given MOI $m$ and observation $\boldsymbol{x}$, the set of all such infections is denoted by

$$M_{\boldsymbol{x}}^{(m)} := \{\boldsymbol{m} \mid \boldsymbol{m} \to \boldsymbol{x}, |\boldsymbol{m}| = m\}. \tag{3a}$$

Furthermore, for an observation $\boldsymbol{x}$, we define the set of all haplotypes that could potentially be present in $\boldsymbol{x}$ as

$$A_{\boldsymbol{x}} := \{\boldsymbol{h} = (h_1, h_2) \mid h_1 \in x_1, h_2 \in x_2\}. \tag{3b}$$

The set of all observations with at most the same alleles detected at each locus as in $\boldsymbol{x}$ is denoted by

$$\mathcal{A}_{\boldsymbol{x}} := \{\boldsymbol{y} = (y_1, y_2) \mid y_1 \subseteq x_1, y_2 \subseteq x_2\}. \tag{3c}$$

We define the partial order "$\leq$" on the set of all possible observations, such that $\boldsymbol{y} \leq \boldsymbol{x}$ is equivalent to $\boldsymbol{y} \in \mathcal{A}_{\boldsymbol{x}}$. If $\boldsymbol{y} \leq \boldsymbol{x}$ and $\boldsymbol{x} \neq \boldsymbol{y}$ we write $\boldsymbol{y} \prec \boldsymbol{x}$. Therefore, we can express $M_{\boldsymbol{x}}^{(m)}$ as

$$M_{\boldsymbol{x}}^{(m)} = \{\boldsymbol{m} \mid m_{\boldsymbol{h}} = 0 \text{ if } \boldsymbol{h} \notin A_{\boldsymbol{x}}, |\boldsymbol{m}| = m\} \setminus \bigcup_{\boldsymbol{y} \prec \boldsymbol{x}} \{\boldsymbol{m} \mid m_{\boldsymbol{h}} = 0 \text{ if } \boldsymbol{h} \notin A_{\boldsymbol{y}}, |\boldsymbol{m}| = m\}, \tag{4}$$

$A_{\boldsymbol{y}}$ is defined as in (3b) for observation $\boldsymbol{y}$ [19]. Consider a host infected $m$ times (MOI= $m$), the probability that the actual infection is $\boldsymbol{x}$, is given by

$$P(\boldsymbol{x} \mid m) = \frac{P(\boldsymbol{x}, m)}{\kappa_m}. \tag{5a}$$

Hence,

$$P(\boldsymbol{x}, m) = P\left(M_{\boldsymbol{x}}^{(m)}\right) = \sum_{\boldsymbol{m} \in M_{\boldsymbol{x}}^{(m)}} P(\boldsymbol{m}) = \sum_{\boldsymbol{m} \in M_{\boldsymbol{x}}^{(m)}} P(\boldsymbol{m}|m) \kappa_m = \kappa_m \sum_{\boldsymbol{m} \in M_{\boldsymbol{x}}^{(m)}} \binom{m}{\boldsymbol{m}} \boldsymbol{p}^{\boldsymbol{m}}. \tag{5b}$$

The probability of observation $\boldsymbol{x}$ becomes

$$P(\boldsymbol{x}) = \sum_{m=1}^{\infty} P(\boldsymbol{x}, m) = \sum_{m=1}^{\infty} \kappa_m \sum_{\boldsymbol{m} \in M_{\boldsymbol{x}}^{(m)}} \binom{m}{\boldsymbol{m}} \boldsymbol{p}^{\boldsymbol{m}}. \tag{5c}$$

The simplified notation $P_{\boldsymbol{x}}$ will be used from now on, in place of $P(\boldsymbol{x})$ to denote the probability of observing $\boldsymbol{x}$. Furthermore, let $|\boldsymbol{x}|$ denote the cardinality of the set $\boldsymbol{x}$. By using (3a) and the inclusion-exclusion principle, the inner sum on the right-hand side of (5c) can be

rewritten [19] as

$$
\begin{aligned}
\sum_{\boldsymbol{m}\in M_{\boldsymbol{x}}^{(m)}} \binom{m}{\boldsymbol{m}}\boldsymbol{p^m} &= \sum_{\substack{\boldsymbol{m}:|\boldsymbol{m}|=m \\ m_{\boldsymbol{h}}=0 \text{ if } \boldsymbol{h}\notin A_{\boldsymbol{x}}}} \binom{m}{\boldsymbol{m}}\boldsymbol{p^m} + \sum_{\boldsymbol{y}\prec\boldsymbol{x}}(-1)^{|\boldsymbol{x}|-|\boldsymbol{y}|}\sum_{\substack{\boldsymbol{m}:|\boldsymbol{m}|=m \\ m_{\boldsymbol{h}}=0 \text{ if } \boldsymbol{h}\notin A_{\boldsymbol{y}}}} \binom{m}{\boldsymbol{m}}\boldsymbol{p^m} \\
&= \sum_{\boldsymbol{y}\preceq\boldsymbol{x}}(-1)^{|\boldsymbol{x}|-|\boldsymbol{y}|}\sum_{\substack{\boldsymbol{m}:|\boldsymbol{m}|=m \\ m_{\boldsymbol{h}}=0 \text{ if } \boldsymbol{h}\notin A_{\boldsymbol{y}}}} \binom{m}{\boldsymbol{m}}\boldsymbol{p^m} \\
&= \sum_{\boldsymbol{y}\in\mathcal{A}_{\boldsymbol{x}}}(-1)^{|\boldsymbol{x}|-|\boldsymbol{y}|}\sum_{\substack{\boldsymbol{m}:|\boldsymbol{m}|=m \\ m_{\boldsymbol{h}}=0 \text{ if } \boldsymbol{h}\notin A_{\boldsymbol{y}}}} \binom{m}{\boldsymbol{m}}\boldsymbol{p^m}.
\end{aligned}
\tag{6a}
$$

Therefore, the probability of observing $\boldsymbol{x}$ in (5c) becomes

$$
P_{\boldsymbol{x}} = \sum_{m=1}^{\infty}\kappa_m\sum_{\boldsymbol{y}\in\mathcal{A}_{\boldsymbol{x}}}(-1)^{|\boldsymbol{x}|-|\boldsymbol{y}|}\sum_{\substack{\boldsymbol{m}:|\boldsymbol{m}|=m \\ m_{\boldsymbol{h}}=0 \text{ if } \boldsymbol{h}\notin A_{\boldsymbol{y}}}} \binom{m}{\boldsymbol{m}}\boldsymbol{p^m}.
\tag{7a}
$$

By the multinomial theorem, we have

$$
\sum_{\substack{\boldsymbol{m}:|\boldsymbol{m}|=m \\ m_{\boldsymbol{h}}=0 \text{ if } \boldsymbol{h}\notin A_{\boldsymbol{y}}}} \binom{m}{\boldsymbol{m}}\boldsymbol{p^m} = \Big(\sum_{\boldsymbol{h}\in A_{\boldsymbol{y}}} p_{\boldsymbol{h}}\Big)^m.
\tag{7b}
$$

By changing the order of summation and using the PGF (7a) becomes

$$
\begin{aligned}
P_{\boldsymbol{x}} &= \sum_{m=1}^{\infty}\kappa_m\sum_{\boldsymbol{y}\in\mathcal{A}_{\boldsymbol{x}}}(-1)^{|\boldsymbol{x}|-|\boldsymbol{y}|}\Big(\sum_{\boldsymbol{h}\in A_{\boldsymbol{y}}} p_{\boldsymbol{h}}\Big)^m \\
&= \sum_{\boldsymbol{y}\in\mathcal{A}_{\boldsymbol{x}}}(-1)^{|\boldsymbol{x}|-|\boldsymbol{y}|}\sum_{m=1}^{\infty}\kappa_m\Big(\sum_{\boldsymbol{h}\in A_{\boldsymbol{y}}} p_{\boldsymbol{h}}\Big)^m \\
&= \sum_{\boldsymbol{y}\in\mathcal{A}_{\boldsymbol{x}}}(-1)^{|\boldsymbol{x}|-|\boldsymbol{y}|}G\Big(\sum_{\boldsymbol{h}\in A_{\boldsymbol{y}}} p_{\boldsymbol{h}}\Big).
\end{aligned}
\tag{7c}
$$

Note, the above equation is correct (cf. eq. 7c) for any distribution of MOI. If we impose the condition Poisson distribution, the MOI parameter $\lambda$ occurs in the PGF. Therefore, the probability $P_{\boldsymbol{x}}$ depends on the MOI parameter $\lambda$ and the haplotype frequencies $\boldsymbol{p}$. The parameter space of the conditional Poisson model is

$$
\Theta := \mathbb{R}^+ \times \mathcal{S}_H = \Big\{(\lambda,\boldsymbol{p}) \,|\, \lambda > 0 \text{ and } \boldsymbol{p}\in\mathcal{S}_H\Big\},
\tag{8}
$$

where, $\mathcal{S}_H := \Big\{(p_1,\dots,p_H)\,\Big|\,\sum_{k=1}^{H} p_k = 1 \text{ and } p_k \geq 0,\text{ for all } k\Big\}$ is the $H-1$-dimensional simplex.

The true parameters, here subsumed by the vector $\boldsymbol{\theta} = (\lambda,\boldsymbol{p})$, are unknown and can be inferred from empirical data. Assume a dataset $\mathcal{X}$ consisting of $N$ observations $\boldsymbol{x}^{(1)},\dots,\boldsymbol{x}^{(N)}$, where the notation $\boldsymbol{x}^{(j)} = \big(x_1^{(j)},x_2^{(j)}\big)$ is used for the $j$th observation. For the dataset $\mathcal{X}$, let $n_{\boldsymbol{x}}$ be the number of times observation $\boldsymbol{x}$ is made. Naturally,

$$
\sum_{\boldsymbol{x}\in\mathcal{O}} n_{\boldsymbol{x}} = N.
$$

Using (7c), the likelihood function of the parameter $\boldsymbol{\theta} = (\lambda, \boldsymbol{p})$ given the data $\mathcal{X}$ is given by

$$\mathcal{L}_{\mathcal{X}}(\boldsymbol{\theta}) = \prod_{j=1}^{N} P_{\boldsymbol{x}^{(j)}} = \prod_{\boldsymbol{x} \in \mathcal{O}} \left( \sum_{\boldsymbol{y} \in A_{\boldsymbol{x}}} (-1)^{|\boldsymbol{x}|-|\boldsymbol{y}|} G\left( \sum_{\boldsymbol{h} \in A_{\boldsymbol{y}}} p_{\boldsymbol{h}} \right) \right)^{n_{\boldsymbol{x}}}. \tag{9}$$

Hence, the log-likelihood function becomes

$$\ell_{\mathcal{X}}(\boldsymbol{\theta}) = \log\left(\mathcal{L}_{\mathcal{X}}(\boldsymbol{\theta})\right) = \sum_{\boldsymbol{x} \in \mathcal{O}} n_{\boldsymbol{x}} \log\left( \sum_{\boldsymbol{y} \in A_{\boldsymbol{x}}} (-1)^{|\boldsymbol{x}|-|\boldsymbol{y}|} G\left( \sum_{\boldsymbol{h} \in A_{\boldsymbol{y}}} p_{\boldsymbol{h}} \right) \right). \tag{10}$$

To obtain the maximum-likelihood estimate (MLE) $\hat{\boldsymbol{\theta}} = (\hat{\lambda}, \hat{\boldsymbol{p}})$ the log-likelihood function needs to be maximized. The complexity of the log-likelihood function does not permit a closed solution, and must be maximized numerically. For this purpose the expectation-maximization (EM)-algorithm will be used [26]. This will be discussed in The maximum-likelihood estimate.

Note that (10) holds for any distribution of MOI. If the positive Poisson distribution is replaced, only the PGF $G$ and the parameter $\boldsymbol{\theta}$ need to be modified.

**Assessing bias and variance of the estimator**  MLEs usually have desirable asymptotic (large sample size) properties. In practice, sample size is often limited, and the quality of the estimator needs to be investigated for small and intermediate sample sizes. Because no explicit solution exists for the MLE, its performance in terms of bias and variance needs to be investigated by numerical simulations. Here, we adapt the approach of [19,27,28].

Bias and variance of the MLE will be affected by: (i) sample size $N$; (ii) the genetic architecture, i.e., the number of alleles $n_1$, $n_2$ segregating at each locus; (iii) the value of the MOI parameter $\lambda$; (iv) the frequency distribution of haplotypes $\boldsymbol{p}$.

To investigate the properties of the MLE for a representative range of parameters we proceeded as follows (parameters used in the simulation study are described below and summarized in Table 1). For a set of parameters $(N, n_1, n_2, \lambda, \boldsymbol{p})$ we generated $K = 100\,000$ datasets $\mathcal{X}_1, \dots, \mathcal{X}_K$ of size $N$ according to the model (7c). For each dataset $\mathcal{X}_k$, the MLE $\hat{\boldsymbol{\theta}}_k = (\hat{\lambda}_k, \hat{\boldsymbol{p}}^{(k)})$ was calculated. From each $\hat{\lambda}_k$ the mean MOI $\hat{\psi}_k$ was calculated according to (1c). The bias and variance of the mean MOI $\psi$ were estimated as

$$\text{bias}(\hat{\psi}) = \overline{\psi} - \psi, \tag{11a}$$

**Table 1. Parameter choice: Summary of model parameters chosen for simulations in the assessment of the performance of the estimator.**

| Parameter | Description | Value | |
|---|---|---|---|
| $K$ | simulated datasets | 100 000 | |
| $(n_1, n_2)$ | allele numbers | (2,2), (4,7) | |
| $N$ | sample size | 50,100,150,200,500 | |
| $\lambda$ | MOI parameter | 0.1,0.25,0.5,1,1.5,2,2.5 | |
| $\boldsymbol{p}$ | haplotype freq. | balanced | unbalanced |
| | $(n_1, n_2) = (2, 2)$ | $p_k = \frac{1}{4}$ | $p_1 = 0.7$, |
| | | $k = 1, \dots, 4$ | $p_k = 0.1, k = 2, \dots, 4$ |
| | $(n_1, n_2) = (4, 7)$ | $p_k = \frac{1}{28}$ | $p_1 = 0.7$, |
| | | $k = 1, \dots, 28$ | $p_k = \frac{0.1}{9}, k = 2, \dots, 28$ |

and

$$\text{Var}(\hat{\psi}) = \frac{1}{K-1} \sum_{k=1}^{K} (\hat{\psi}_k - \bar{\psi})^2, \tag{11b}$$

where

$$\bar{\psi} = \frac{1}{K} \sum_{k=1}^{K} \hat{\psi}_k. \tag{11c}$$

To allow comparisons between different parameter ranges it is more appropriate to consider the relative bias and coefficient of variation, which are independent of the scale, i.e.,

$$\frac{\text{bias}(\hat{\psi})}{\psi}, \tag{12a}$$

and

$$\frac{\sqrt{\text{Var}(\hat{\psi})}}{\psi}. \tag{12b}$$

For each haplotype frequency $p_h$, bias and variance were defined in the same way with obvious modifications.

*Genetic architecture*

Considering the number of alleles at each of the 2 loci considered, to investigate the performance of the estimator, we assumed the cases, (i) $n_1 = n_2 = 2$ and (ii) $n_1 = 4, n_2 = 7$. This yields, respectively, 4 and 28 possible haplotypes in total. The simulation is based on those cases, as an exhaustive simulation on all possible configurations would be too expensive computationally due to the curse of dimensionality.

The model is not restricted to the number of alleles per locus. However, $\sim 30$ alleles per locus would result in $\sim 900$ parameters, which would require a sample size of $N > 1\,000$ to obtain reliable results.

*MOI parameter*

Concerning the MOI parameter we chose $\lambda = 0.1, 0.25, 0.5, 1, 1.5, 2, 2.5$, corresponding to a mean MOI $\psi = 1.05, 1.13, 1.27, 1.58, 1.93, 2.31, 2.72$. In the case of malaria, this corresponds to low transmission $\psi < 1.27$, intermediate transmission $1.27 \leq \psi < 1.93$, and high transmission $\psi \geq 1.93$ [28].

*Haplotype frequency distribution*

The following haplotype frequency distributions $\boldsymbol{p}$ were chosen. First, a completely uniform (balanced) distribution was chosen, i.e., each of the $H = n_1 n_2$ haplotype had the same frequency,

$$p_1 = \ldots = p_H = \frac{1}{H}. \tag{13a}$$

Then, an unbalanced distribution with one predominant haplotype was chosen. The frequency of the predominant haplotype was chosen to be 70%, while the remaining haplotypes

all had the same frequency. In particular, we chose

$$p_1 = 0.7, p_2 = \dots = p_H = \frac{0.3}{H-1}. \tag{13b}$$

For the genetic architecture $n_1 = n_2 = 2$ this yielded, $p_1 = 0.7, p_2 = p_3 = p_4 = 0.1$ and $n_1 = 4, n_2 = 7$ yielded $p_1 = 0.7, p_2 = \dots = p_{28} = 0.011$.

*Sample size*

The performance of an estimator is usually affected by sample size. We investigate the effect of sample size in our numerical investigations by constructing datasets of size $N = 50, 100, 150, 200, 500$, which are typical sample sizes ranging from areas of low to high transmission in diseases like malaria.

The simulation study as well as the graphical outputs are implemented in R [29]. The code is available at https://github.com/Maths-against-Malaria/MultiAllelicBiLociModel and on Zenodo at https://doi.org/10.5281/zenodo.8289710.

## Linkage disequilibrium

Linkage disequilibrium (LD) also known as gametic disequilibrium is a measure of the association between alleles at different loci, i.e. statistical dependence. LD is calculated from haplotype frequency estimates. Two loci are said to be in linkage equilibrium if the haplotype frequencies coincide with the products of allele frequencies, i.e., statistical independence, otherwise, they are said to be in LD. For two bi-allelic loci, LD measures have rather straightforward interpretations (cf. [30]). For multi-allelic loci LD measures and their interpretation are less straightforward. There exist a variety of multi-allelic LD measures in the literature, each having its advantages and limitations (cf. [22]). Here, to assess the quality of the MLEs when used as plug in estimates to derive LD, we focus on three commonly used measures, i.e., $D'$, $r^2$, and $Q^*$.

We further exemplify the three LD measures with an empirical data application. In the data application, we also derive the more recently developed asymmetric conditional LD (ALD) measure introduced in [23], which accounts for asymmetry in the number of alleles observed at each locus.

The various LD measures are defined as follows. Let $x_i$ ($0 < x_i < 1$) denote the frequency of allele $A_i$ ($i = 1, \dots, n_1$) at the first and $y_j$ ($0 < y_j < 1$) of allele $B_j$ ($j = 1, \dots, n_2$) at the second locus. The loci are referred to as locus $A$ and $B$, respectively. All LD measures are undefined if at least one of the loci is monomorphic. The measure $D'$ defined in [22] is calculated as

$$D' := \sum_{i=1}^{n_1} \sum_{j=1}^{n_2} x_i y_j \frac{|D_{ij}|}{D_{\max}}, \tag{14a}$$

where $D_{ij} = p_{ij} - x_i y_j$ is the difference between the observed frequency of haplotype $A_i B_j$ and the expected frequency $x_i y_j$ of $A_i B_j$ assuming a random association of the alleles $A_i$ and $B_j$, and

$$D_{\max} := \begin{cases} \min\left(x_i y_j, (1-x_i)(1-y_j)\right) & \text{if } D_{ij} < 0 \\ \min\left(x_i(1-y_j), y_j(1-x_i)\right) & \text{if } D_{ij} > 0. \end{cases}$$

The measure $r^2$ also known as $D^*$ [22] is defined by

$$r^2 := \frac{1}{H_A H_B} \sum_{i=1}^{n_1} \sum_{j=1}^{n_2} D_{ij}^2, \tag{14b}$$

where $H_A = 1 - \sum_{i=1}^{n_1} x_i^2$ and $H_B = 1 - \sum_{j=1}^{n_2} y_j^2$ are the Hardy-Weinberg heterozygosities at loci $A$ and $B$, respectively. Finally, the measure $Q^*$ is defined in [22] as

$$Q^* := \frac{1}{(n_1 - 1)(n_2 - 1)} \sum_{i=1}^{n_1} \sum_{j=1}^{n_2} \frac{D_{ij}^2}{x_i y_j}. \tag{14c}$$

Note, $r^2$ and $Q^*$ are equivalent for bi-allelic loci, i.e., $n_1 = n_2 = 2$.

Moreover, [23] defines ALD between the two loci conditioned on $A$ by

$$W_{B|A} := \sqrt{\frac{1}{H_B} \sum_{i=1}^{n_1} \sum_{j=1}^{n_2} \frac{D_{ij}^2}{x_i}}, \tag{14d}$$

while ALD between loci $A$ and $B$ conditioned on $B$ is defined by

$$W_{A|B} := \sqrt{\frac{1}{H_A} \sum_{i=1}^{n_1} \sum_{j=1}^{n_2} \frac{D_{ij}^2}{y_j}}. \tag{14e}$$

For bi-allelic loci (14d) and (14e) equal and coincide with the square root of $r^2$, i.e., $W_{B|A}^2 = W_{A|B}^2 = r^2$.

## Results

Estimating pathogen haplotype frequencies is a fundamental basic to monitor, for instance, haplotypes of interest (e.g. drug-resistance-associated haplotypes) or to reconstruct past or ongoing evolutionary events such as selection. The latter is typically investigated by considering summary statistics such as pairwise linkage disequilibria (LD) [22,31], which are calculated from haplotype frequencies.

Assuming a genetic architecture of two multi-allelic loci, we first show how the MLE is derived. Then we provide an application to empirical data and contrast different LD measures. Finally, we investigate the finite sample properties of the estimator.

### The maximum-likelihood estimate

The estimates for haplotypes frequencies and MOI are obtained by maximizing the likelihood function (10). Because the likelihood function (10) is high-dimensional and non-linear, no closed solution could be found. Therefore, the likelihood function has to be maximized numerically. Applying Newton or Quasi-Newton methods to maximize the likelihood function over a multi-dimensional space (particularly over the simplex as in our case) is inconvenient in practice because convergence is too sensitive to initial conditions. Therefore, we pursue with the expectation maximization (EM)-algorithm. The algorithm is typically insensitive to initial conditions but tends to be slow close to the point of convergence (cf. [32] Chapter 2).

The EM-algorithm is a two-step recursive method consisting of (i) the expectation-step (E) and (ii) the maximization-step (M). Starting from an initial parameter choice $\boldsymbol{\theta}_0$, it updates the parameter vector $\boldsymbol{\theta}_t$ in every step $t$ until convergence is reached numerically, to yield the MLE. First the log-likelihood of the unobserved MOI, i.e., $(m, \boldsymbol{m})$ which are random variables, and the unknown parameters $\boldsymbol{\theta}$, given the observed data is calculated. At iteration $t$, in the E-step, the expectation of this likelihood function is calculated with respect to the distribution of the unobserved random variable, given the observed data and the parameter choice $\boldsymbol{\theta}_t$. In the M-step the resulting function is maximized with respect to the unknown parameters $\boldsymbol{\theta}$, which yields the updated parameters vector $\boldsymbol{\theta}_{t+1}$. We derive the algorithm in section The EM-algorithm using a plugin estimate for the MOI parameter of the S1 Mathematical appendix in detail.

In the present case, the EM-algorithm yields the following iteration. First, arbitrary initial values of the MOI (Poisson) parameter $\lambda^{(0)}$ and haplotype frequencies $\boldsymbol{p}^{(0)}$ are chosen. First, in step $t + 1$, the frequency estimate of each haplotype $\boldsymbol{h}$ is updated as

$$p_{\boldsymbol{h}}^{(t+1)} = \frac{C_{\boldsymbol{h}}^{(t)}}{\sum\limits_{\boldsymbol{h} \in \mathcal{H}} C_{\boldsymbol{h}}^{(t)}}, \tag{15a}$$

where

$$C_{\boldsymbol{h}}^{(t)} = p_{\boldsymbol{h}}^{(t)} \sum_{\boldsymbol{x} \in \mathcal{O}} n_{\boldsymbol{x}} \frac{\sum\limits_{\boldsymbol{y} \in \mathcal{A}_{\boldsymbol{x}}} (-1)^{|\boldsymbol{x}| - |\boldsymbol{y}|} G_t'\left(\sum\limits_{i \in A_{\boldsymbol{y}}} p_i^{(t)}\right) I_{A_{\boldsymbol{y}}}(\boldsymbol{h})}{\sum\limits_{\boldsymbol{y} \in \mathcal{A}_{\boldsymbol{x}}} (-1)^{|\boldsymbol{x}| - |\boldsymbol{y}|} G_t\left(\sum\limits_{i \in A_{\boldsymbol{y}}} p_i^{(t)}\right)}, \tag{15b}$$

$$I_{A_{\boldsymbol{y}}}(\boldsymbol{h}) = \begin{cases} 1 & \text{if } \boldsymbol{h} \in Ay, \\ 0 & \text{if } \boldsymbol{h} \notin Ay. \end{cases} \tag{15c}$$

and $G_t(z)$ is the PGF (1b) with parameters in step $t$. (Note that eq. 15 is not restricted to the conditional Poisson distribution.)

At the second step of the algorithm, the MOI parameter $\lambda_{t+1}$ is updated by iterating the equation

$$x_{\tau+1} = x_\tau - \frac{x_\tau - \dfrac{B^{(t)}}{N}\left(1 - e^{-x_\tau}\right)}{1 + x_\tau - \dfrac{x_\tau}{1 - e^{-x_\tau}}}, \tag{15d}$$

where

$$B^{(t)} = \sum_{\boldsymbol{x} \in \mathcal{O}} n_{\boldsymbol{x}} \frac{\sum\limits_{\boldsymbol{y} \in \mathcal{A}_{\boldsymbol{x}}} (-1)^{|\boldsymbol{x}| - |\boldsymbol{y}|} \sum\limits_{\boldsymbol{h} \in A_{\boldsymbol{y}}} p_{\boldsymbol{h}}^{(t)} G'\left(\sum\limits_{i \in A_{\boldsymbol{y}}} p_i^{(t)}\right)}{\sum\limits_{\boldsymbol{y} \in \mathcal{A}_{\boldsymbol{x}}} (-1)^{|\boldsymbol{x}| - |\boldsymbol{y}|} G_t\left(\sum\limits_{i \in A_{\boldsymbol{y}}} p_i^{(t)}\right)}, \tag{15e}$$

starting from $x_0 = \lambda_t$ until convergence is reached numerically. Convergence is reached if $|x_{\tau+1} - x_\tau| < \varepsilon$ holds. The MOI parameter is updated as $\lambda_{t+1} = x_{\tau+1}$.

These two steps are repeated until convergence is reached numerically. More precisely, the algorithm terminates at step $t+1$ if $|\lambda_{t+1} - \lambda_t| + \|p_{\boldsymbol{h}}^{(t+1)} - p_{\boldsymbol{h}}^{(t)}\|_2 < \varepsilon$ holds. The MLE is then given by

$$\hat{p}_{\boldsymbol{h}} = p_{\boldsymbol{h}}^{(t+1)} \quad \text{and} \quad \hat{\lambda} = \lambda_{t+1}. \tag{16}$$

Depending on the application, the haplotype-frequency estimation is of primary interest, while there is prior information concerning MOI. The EM-algorithm can be easily modified to allow the possibility of inputting a plugin estimate of the MOI parameter. The EM-algorithm can be adapted to this case following [19]. This is described in section The EM-algorithm using a plugin estimate for the MOI parameter in S1 Mathematical appendix.

The EM-algorithm (with and without a plugin estimate for the MOI parameter) is implemented as an R script available in the supplementary material, also on GitHub at https://github.com/Maths-against-Malaria/MultiAllelicBiLociModel, and on Zenodo at https://doi.org/10.5281/zenodo.8289710. Importantly, the implementation also allows deriving bootstrap-percentile confidence intervals (CIs) and bootstrap bias corrections of the estimates (cf. [33]) for the MLEs. A detailed description of the usage of this script is provided in S1 User Manual.

## Data application

We estimate linkage disequilibrium (LD) from molecular data of malaria parasites extracted from blood samples, which were positive for *P. falciparum*, collected in Yaoundé, Cameroon from 2001 to 2002 ($N = 166$ samples) and 2004 to 2005 ($N = 165$ samples). The dataset was previously described in [11]. The data contains information associated with sulfadoxine--pyrimethamine (SP) resistance [34–36]. Particularly, allelic information is available at (i) codons 51, 59, 108, and 164 at *Pfdhfr*, (ii) codons 436, 437, 540, 581, and 613 at *Pfdhps*, (iii) 18 STR markers on chromosome 4 flanking *Pfdhfr*, (iv) 15 STR markers on chromosome 8 flanking *Pfdhps*, and (v) 8 neutral STR markers at chromosome 2 and 3 [11]. Because no mutations occurred on codon 164 at *Pfdhfr* and codons 540 and 581 at *Pfdhps*, they were excluded from pairwise LD analysis. The remaining six SNPs were all bi-allelic.

Pairwise LD was measured by $D'$, $r^2$, $Q^*$, and ALD. We estimated pairwise LD for (i) the 6 SNPs retained at *Pfdhfr* and *Pfdhps*, and (ii) all the 41 STR markers. Reported CIs are bootstrap percentile CIs [33].

Note that the ready-to-use model implementation provided as R script in supplement provides several functions to derive MLEs and LD estimates from a pair of multi-allelic markers. More precisely, the function "mle(<data>,...)" allow deriving estimates of haplotype frequencies and MOI, with <DATA> being the data in a special format, and "..." representing extra argument allowing to derive the MLEs with bootstrap confidence intervals, bootstrap bias-corrected estimates, or both (see S1 User Manual for more details). In particular, the following code is executed to derive MLEs from a dataset ("DATA") containing a pair of multi-allelic markers with genetic architecture $n_1 = 2, n_2 = 3$ (argument "c(2,3)"), alongside the 95% bootstrap confidence interval ("CI = TRUE") with 15 000 bootstrap repeats:

```
mle(DATA, c( 2 , 3 ), CI = TRUE, B = 15000)
```

The generated output is:

```
## $lambda
##                        2.5%       97.5%
## 0.14669947 0.04858829 0.27565340
##
## $p
##                        2.5%       97.5%
## 00 0.1479238 0.08471595 0.2191917
## 01 0.1479238 0.08067984 0.2219839
## 02 0.2042203 0.12645067 0.2850759
## 10 0.1820436 0.11044417 0.2576321
## 11 0.1820436 0.10990826 0.2604331
## 12 0.1358449 0.07137319 0.2074471
##
## $haplotypes
##      [,1] [,2]
## [1,]    0    0
## [2,]    0    1
## [3,]    0    2
## [4,]    1    0
## [5,]    1    1
## [6,]    1    2
##
## $`Sample size`
## [1] 100
```

Estimates of LD, i.e., $D'$, $r^2$, $Q^*$, and the ALD measures $W_{A|B}$ and $W_{B|A}$ are obtained with the function "ld(<DATA>,…)" (see S1 User Manual for more details).

For the same data as above, estimates of $D'$, $r^2$, $Q^*$, and the ALD measures $W_{A|B}$ and $W_{B|A}$, alongside their 95% confidence interval with 15 000 replicates are obtained by running the following code:

```
ld(DATA, c(2,3), CI = TRUE)
```

which generates the output:

```
##                      2.5%        97.5%
## D'        0.1366585  0.03311191  0.3384459
## r^2       0.01050607 0.0006702836 0.06949526
## Q*        0.01040208 0.0006640142 0.06714261
## W[A | B]  0.1024991  0.02588983  0.2636195
## W[B | A]  0.1447816  0.03673931  0.3732486
```

Fig. 4 shows pairwise LD measured by $D'$ and ALD for the 6 polymorphic SNPs at *Pfdhfr* and *Pfdhps*. In general, LD was high at *Pfdhfr* in the years 2001/2002 (as evidenced by the light regions): $D' = 0.79$ (95% CI: [0.62,0.95]) between codons 51 and 59, $D' = 1$ (95% CI: [1,1]) between codons 51 and 108, and $D' = 0.95$ (95% CI: [0.85,1]) between 59 and 108

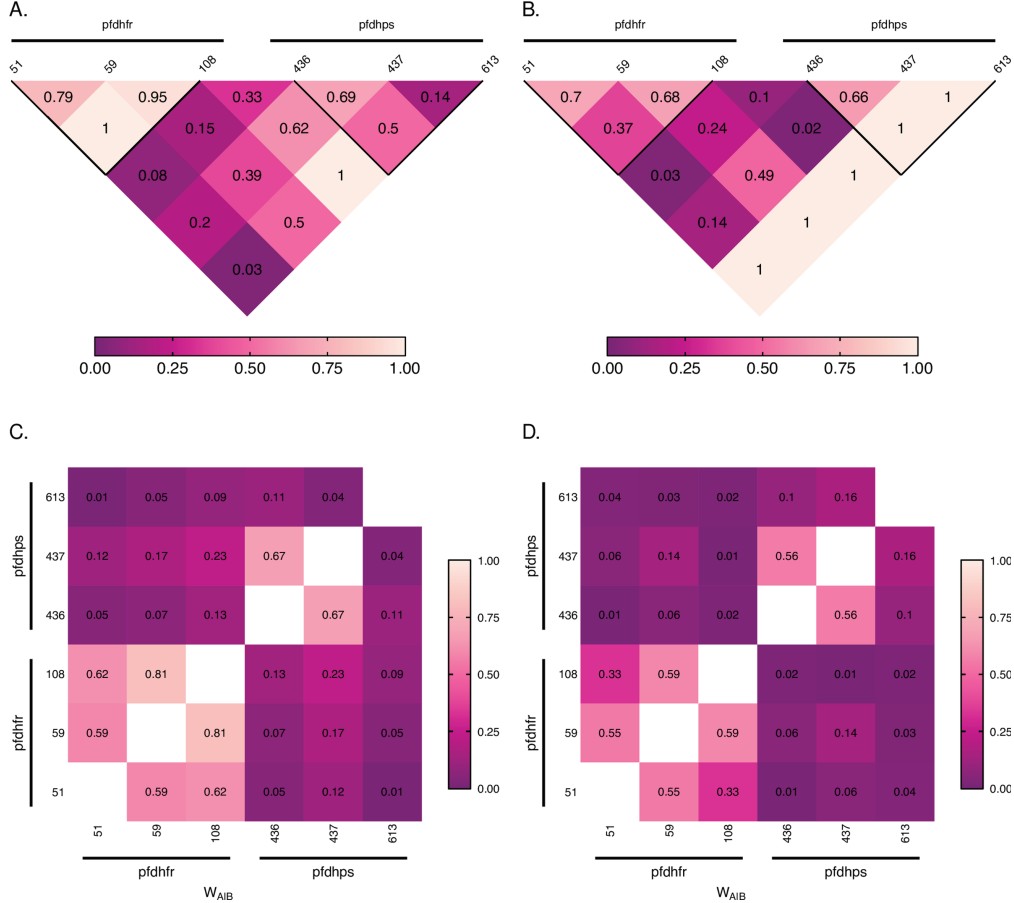

**Fig 4. Linkage disequilibrium at SNP markers:** Shown are estimates of pairwise LD for six SNP marker loci associated with SP drug-resistance, i.e., three codons (51, 59, 108) at *Pfdhfr* and three codons (436, 437, 613) at *Pfdhps*. Panels (A, B) show $D'$ values, whereas (C, D) show ALD values. Furthermore, panels (A, C) correspond to the years 2001/2002, while panels (B, D) correspond to the years 2004/2005. The codons on *Pfdhfr* and *Pfdhps* are highlighted on the maps by horizontal black lines, with the name of the gene and codons are specified on top and below the line, respectively. The thick black lines in panels (A, B) group pairwise LD within each gene. The numbers indicate LD values.

(Fig 4A). This is not surprising since pyrimethamine resistance is acquired sequentially, starting with a base mutation at codon 108, which is followed by a secondary mutation either at codon 51 or 59 to cause intermediate resistance [35]. A combination of the three mutations then causes high levels of resistance [35]. The mutations at *Pfdhps* are associated with sulfadoxine resistance, i.e., with resistance against the slow-acting component [35]. Levels of LD were generally lower (Fig 4A) than at *Pfdhfr*, which coincides with the empirical observations elsewhere (e.g. [20]). High LD values between some SNPs at *Pfdhfr* and *Pfdhps* are indicated by $D'$ (Fig 4A, light shaded areas) and neither by ALD (Fig 4C) nor $r^2$ (coinciding with $Q^*$; S1 Fig), for which the corresponding areas in the plot appear rather dark shaded. This is an artifact of $D'$ which occurs for unbalanced frequency distributions when some haplotypes are absent because of low allele frequencies. The corresponding values of ALD (symmetric in this case) are substantially smaller. The values of $r^2$ ($Q^*$) are the squared ALD values in this case and hence close to zero. It should be noted that multi-allelic LD measures can be notoriously

difficult to interpret and that these are prone to artifacts. Hence, it is advisable to look at several measures. Strong disagreement between measures (as evidenced here by areas that appear as slightly shaded areas in one, but as dark shaded areas in another plot) typically highlight data artifacts.

Overall, these results are indicative of recent and independent spreads of pyrimethamine (*Pfdhfr*) and sulfadoxine (*Pfdhps*) resistance, which were still ongoing in 2001/2002. The results are in agreement with the high level of SP drug pressure during that time when chloroquine was discontinued as first-line treatment and replaced by amodiaquine as first-, and SP as second-line treatment [11]. ALD values (Fig 4C) are in agreement with $D'$. Because $r^2$ and $Q^*$ are the squared ALD measure, they yield values close to zero (S1 Fig).

The LD map for the years 2004/2005 reveals a decrease of pairwise LD between codons at *Pfdhfr*, i.e., $D' = 0.7$ (95% CI: [0.42,1]) between 51 and 59, $D' = 0.37$ (95% CI: [0.12,0.67]) between 51 and 108, and $D' = 0.68$ (95% CI: [0.36,1]) between 59 and 108 (see Fig 4B).

The reason for the decrease is the independent mutational origins at codons 51 and 59 succeeding the mutation at codon 108. While the wildtype and 108 single mutants decreased in frequency, the 51/108 and 59/108 double mutants as well as the 51/59/108 triple mutant increased in 2004/2005. Single mutations at codons 51 and 59 were almost absent in 2001/2002, but the 51/108, 59/108 double mutants, and the 51/59/108 triple mutant were abundant. This led to a strong LD in 2001/2002. However, in 2004/2005 the 51 and 59 single mutations were rarely observed. Because of the overall very unbalanced haplotype frequencies, this led to a particularly strong drop in LD between codons 51 and 108. These results have to be considered with caution, as they are likely sampling artifacts in combination with the particularities of LD measures for highly unbalanced distributions. This is reflected by the wide CIs in 2004/2005.

Looking at the LD map for the microsatellite data (Fig 5), similar patterns are observed in the years 2001/2002 and 2004/2005. High LD is observed around the *Pfdhfr* gene (chromosome 4; Fig 5) as evidenced by the light shaded areas, indicative of relatively strong selection on *Pfdhfr*, being reflected by the high frequency of resistance-associated mutations (cf. [19], Table 3). The theoretical prediction is, that a selective sweep causes LD which vanishes with increasing recombinational distance. Hence, highest LD should be observed around the markers flanking *Pfdhfr*. However, LD is higher at the markers upstream *Pfdhfr* (from -58kb to -0.3kb). Due to the stochastic nature of recombination at early phases of a selective sweep, patterns of selection (LD in this case) are typically asymmetric [37]. This is evidenced by the darker shade of the markers downstream *Pfdhfr* in the triangle corresponding to chromosome 4 in Fig 5). LD is also observed around *Pfdhps*, but to a lesser extent, in agreement with the observation that selection is weaker than for *Pfdhfr* [20]. Moreover, LD is reduced between *Pfdhfr* and *Pfdhps*, as seen by the darker region in Fig 5A,B. However, LD appears suspiciously strong between neutral markers, and between neutral markers and those flanking *Pfdhfr* and *Pfdhps*. This counterintuitive observation is purely an artifact of $D'$ caused by the small sample size. Specifically, a large number of alleles are segregating at the neutral markers. For instance, for $n_1 = 20$ and $n_2 = 25$ alleles, 500 possible haplotypes emerge. Further, assume all 500 possible haplotypes had a frequency of 0.002, such that the two markers are in perfect linkage equilibrium (LE). In a sample of size $N = 165$, it is likely to sample all alleles at both loci, but most haplotypes will not be present. (Namely, each allele at marker 1 has frequency 5%, while each on locus 2 has 4%, such that one expects on average that each allele at marker 1 is present at least in 8 samples, while those on marker 2 are present in more than 6 samples, while any given haplotype is on average no expected to be present is a given sample.) The result is high LD (most of the terms $\frac{|D_{ij}^{(kl)}|}{D_{max}^{(kl)}}$ in (14a) are equal to 1). This is the reason, why $D'$

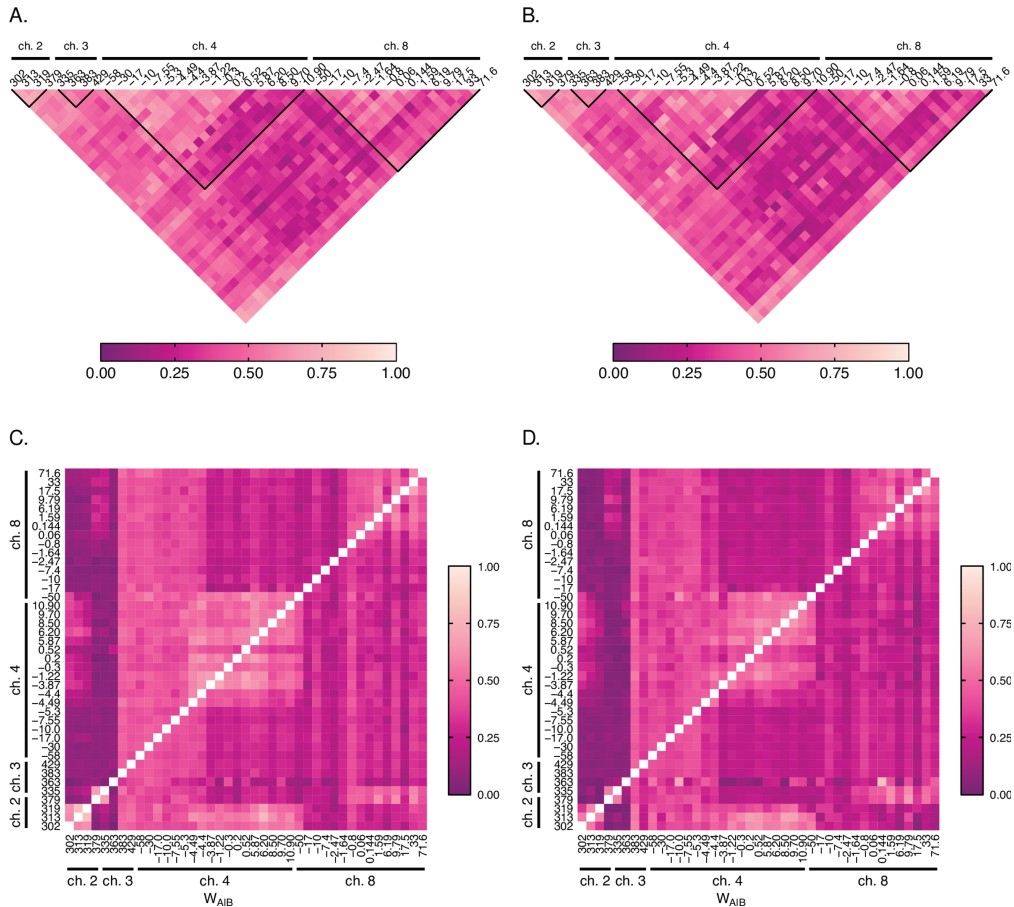

**Fig 5. Linkage disequilibrium at microsatellite markers**: As in Fig 4 but for a set of genes at two neutral chromosomes, i.e., chromosomes 2 and 3, genes around *Pfdhfr* on chromosome 4, and genes around *Pfdhps* on chromosome 8.

overestimates LD for highly polymorphic markers if sample size is only moderate. This artifact does not occur for markers flanking *Pfdhfr* and *Pfdhps* due to the reduced polymorphism subject to genetic hitchhiking. Comparison with ALD, exposes this artifact. Namely, the visible dark stripe at chromosomes 2 and 3 indicates low LD. It is typical for ALD not to be symmetric. High LD would be indicated by low values which occur in a square shape, as observed around *Pfdhfr* and *Pfdhps*.

A second reason for the high LD values is the data quality. At each marker a substantial amount of samples failed to successfully amplify, resulting in large amounts of missing data. I.e., for any pair of markers at chromosome 2 and 3 the number of samples, with missing data at one or both markers is high. This resulted in the artifact that many haplotypes with distinct alleles at both markers were observed, yielding high LD.

## Evaluation of the estimator's performance

Estimates for the MOI parameter and haplotype frequencies are subsumed by $\hat{\theta} = (\hat{\lambda}, \hat{p})$ in the following. Maximum-likelihood estimators typically have desirable asymptotic properties (as proven for a simpler case in [24]). However, the performance of the estimator has to

be investigated for finite sample size. Ultimately, a good estimator is accurate, i.e., low bias, and precise, i.e., low variance. Here, these two components of performance are investigated by numerical simulations because the MLE has no closed form solution.

Together, bias and variance yield the mean squared error (MSE), defined by $\mathrm{MSE}(\hat{\theta}) := \left[\mathrm{Bias}(\hat{\theta})\right]^2 + \mathrm{Var}(\hat{\theta})$. The assessment of an estimator using the MSE implies a trade-off between variance and bias.

**Bias of the estimator**   Here, to facilitate the comparison of bias across different parameter values, we estimate bias as a normalized quantity, namely the relative bias (12a) in percent.

*Relative bias for haplotypes frequencies estimates.*

In the case of a balanced true haplotype distribution, the estimator appears to be unbiased, irrespectively of sample size and number of alleles $n_1$ and $n_2$ considered at each locus (see Fig 6). This result is expected, because for a balanced distribution, the haplotypes are equally represented in the population, making them equivalent. High values of relative bias in this case, are simulation artifact, explained by random sampling, which effect is emphasized for small sample sizes and large number of alleles $n_1$ and $n_2$.

Bias remains low for an unbalanced true haplotype frequency distribution (see Fig 7). Haplotypes with low frequency are under-represented in datasets, hence, are estimated with a higher bias than predominant haplotypes that are over-represented. Therefore, haplotype frequencies will be overestimated for predominant haplotypes and underestimated for the under-represented ones. This is particularly noticeable for high mean MOI ($\psi > 1.8$) and small sample size ($N<150$). A comparison of bias for both the predominant and under-represented haplotypes for $n_1 = n_2 = 2$ (see Figs 7A and B) with the bias of those haplotypes for $n_1 = 4$ and $n_2 = 7$, shows that bias tends to increase with the number of alleles considered. In fact, an increase in $n_1$ and $n_2$ yields a geometrical increase in the number of haplotypes. Therefore, assuming the same frequency of predominant haplotypes, the frequencies of some remaining haplotypes have to decrease for larger values of $n_1$ and $n_2$.

*Relative bias for MOI parameter estimates.*

Estimates for the mean MOI $\psi$ are empirically more relevant than estimates for the MOI parameter $\lambda$. Therefore, we evaluated the bias for $\psi$ instead of $\lambda$. Overall, the estimator has low bias for estimates of mean MOI (see Fig 8). Note that $\lambda$ and therefore $\psi$ have a lower but

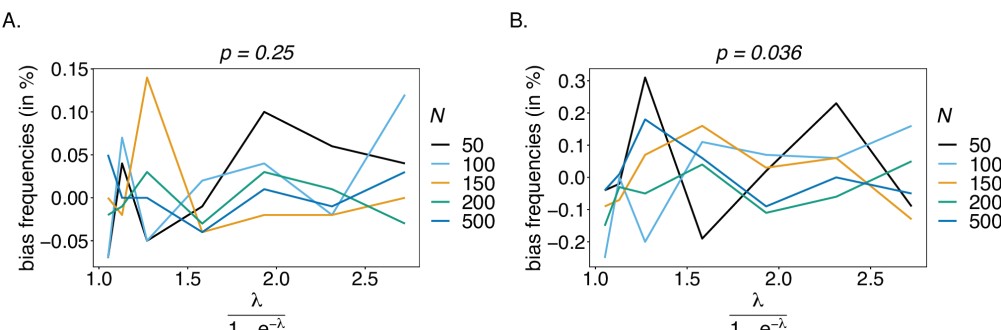

**Fig 6. Bias of frequencies estimates—the balanced case**: Shown is the bias of the frequency estimates in % of the true value, as a function of the mean MOI (i.e., for a range of MOI parameters). The balanced haplotype frequency distributions given in Table 1 for $n_1 = n_2 = 2$ (A) and for $n_1 = 4$, $n_2 = 7$ (B). In both panels, only the bias for the first haplotype is shown (in the balanced case, all haplotypes are equivalent). Colors correspond to different sample sizes.

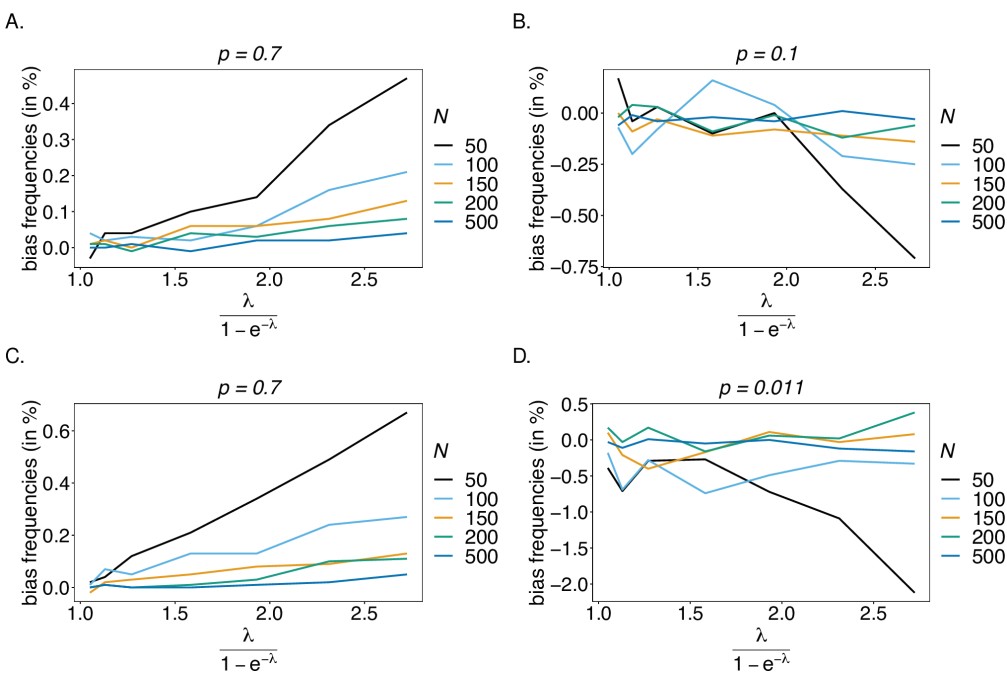

**Fig 7. Bias of frequencies estimates – the unbalanced case:** Shown is the bias of the frequency estimates in % of the true value, as a function of the mean MOI (i.e., for a range of MOI parameters). The unbalanced haplotype frequency distributions given in Table 1 for $n_1 = n_2 = 2$ are assumed in (A, B), while those for $n_1 = 4$, $n_2 = 7$ are assumed in (C, D). In both cases, only the bias for the predominant haplotype and one underrepresented haplotype are shown (all underrepresented haplotypes are equivalent). Colors correspond to different sample sizes.

no upper bound. Hence, they can be overestimated by an unlimited amount, but not arbitrarily underestimated. For larger $n_1$ and $n_2$, and small sample sizes, rare super-infections with many different haplotypes are occasionally over-represented, yielding a substantial overestimation of $\lambda$ and $\psi$. Therefore, bias tends to increase with $\psi$ and decrease with sample size. For unbalanced true haplotype distributions, rare haplotypes are under-represented in datasets, and – particularly for large $\psi$ – single infections with the rare haplotypes are unlikely, which yields a higher bias (see Figs 8C, D).

**Variance of the estimator**   The variance is only reported for the mean MOI. The estimator's variance is evaluated across a range of parameter values, which requires to normalize variance for better comparison. Hence, the estimator's variance was assessed using the coefficient of variation (12b).

In general, the variance of the estimator is small. A comparison between the case $n_1 = n_2 = 2$ and $n_1 = 4$, $n_2 = 7$ shows that variance increases with the number of alleles present at each locus (compare Figs 9A, B and C, D). Due to the geometrical increase of the number of haplotypes as $n_1$ and $n_2$ increase, the true underlying population is less accurately represented. As expected, variance decreases with sample size as the dataset is more representative of the true underlying population.

**Linkage disequilibrium**   We estimated the quality of the MLEs as inputs for the estimation of linkage disequilibrium. While we do not have any preference for any of the measures

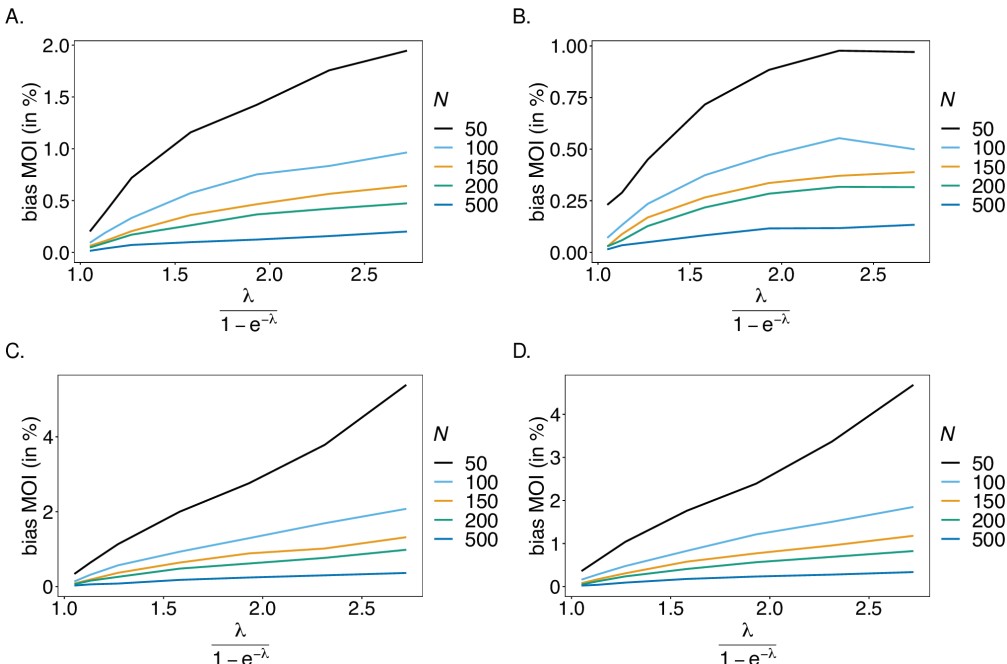

**Fig 8. Bias of MOI estimates**: Shown is the bias of the mean MOI estimates $\psi$ in % as a function of the true mean MOI (i.e., for a range of MOI parameters). Balanced haplotype frequency distributions (Table 1) are assumed for $n_1 = n_2 = 2$ (A) and $n_1 = 4$, $n_2 = 7$ (B), whereas unbalanced distributions are assumed in panels (C) and (D), for $n_1 = n_2 = 2$ and $n_1 = 4$, $n_2 = 7$, respectively. Colors correspond to different sample sizes.

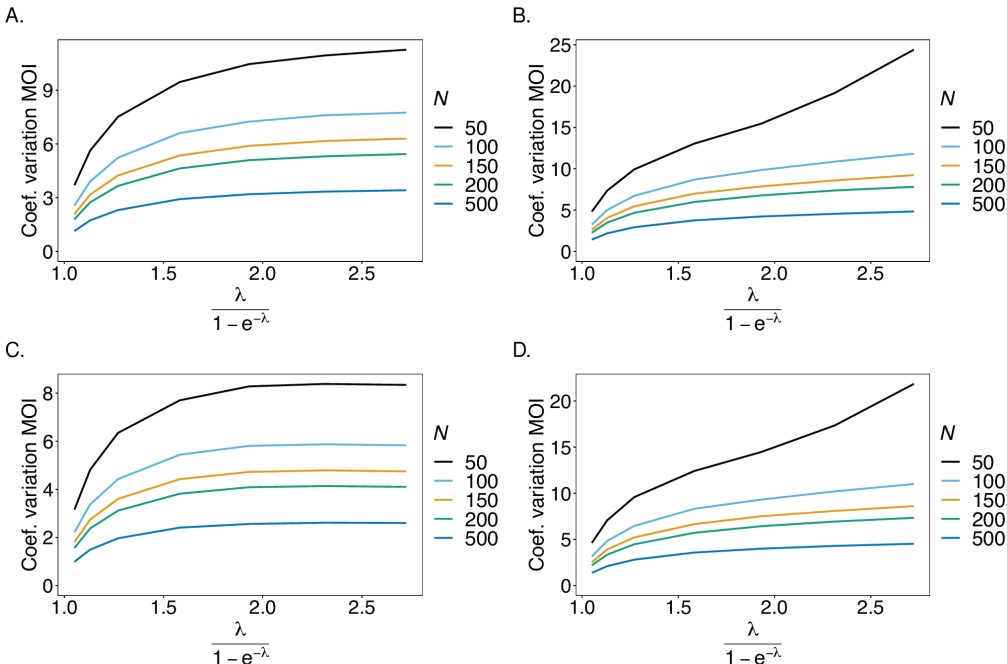

**Fig 9. Variance of MOI estimates**: Shown is the coefficient of variation (CV) of the mean MOI estimates $\psi$ in % as a function of the true mean MOI (i.e., for a range of MOI parameters). Concerning the genetic architecture $n_1 = n_2 = 2$ is assumed in (A, C) and $n_1 = 4$, $n_2 = 7$ in (B, D). The balanced haplotype frequency distributions of Table 1 are assumed in (A, B) and the unbalanced ones in (C, D). Colors correspond to different sample sizes.

of LD, we focused on three measures, namely, $D'$, $r^2$, and $Q^*$ (see eqs.14a, 14b, 14c, respectively). Note that, for $n_1 = n_2 = 2$, the measures $r^2$ and $Q^*$ are equivalent. In the case of a balanced true haplotype frequency distribution, all haplotypes are equally distributed in the population. Therefore, the frequencies are in linkage equilibrium (LE), i.e., $D' = r^2 = Q^* = 0$. In general, the estimates of LD obtained from the MLEs are close to their expected values (see Fig 10), however, they cannot be exactly zero, because all LD measures are non-negative. If the frequencies are in LE, in relative terms, the estimates of LD using $D'$ depart from LE, more than those based on $r^2$ and $Q^*$ (see Fig 10). This is explained by the fact that, $D'$ depends more on the values of haplotype frequencies than $r^2$ and $Q^*$ [22]. A comparison between Figs 10A, B, and C, D shows that the error is larger for larger values of $n_1$ and $n_2$. This result is a reflection of the larger bias of the estimator due to the increased number of haplotypes for larger $n_1$ and $n_2$. Importantly, the error is reduced as sample size increases.

Assuming an unbalanced true haplotype frequency distribution, the estimates of LD are close to the true values (see Fig 11). The haplotype distributions are no longer in LE. In this case, the estimates of LD based on $D'$ depart less from their true values than for balanced true haplotype frequency distributions (compare Figs 10 and 11). This is because, in the unbalanced distribution, one of the haplotypes is predominant with a high frequency, while the remaining haplotypes are rare. Therefore, the estimated value of $D'$ mostly depends on the estimated frequency of the predominant haplotype, which is estimated with low bias (see Fig 6). The deviations from the true value decrease with increasing sample size. The deviations of $r^2$ and $Q^*$ from their true values are slightly higher for unbalanced frequency distributions, but still smaller than for $D'$.

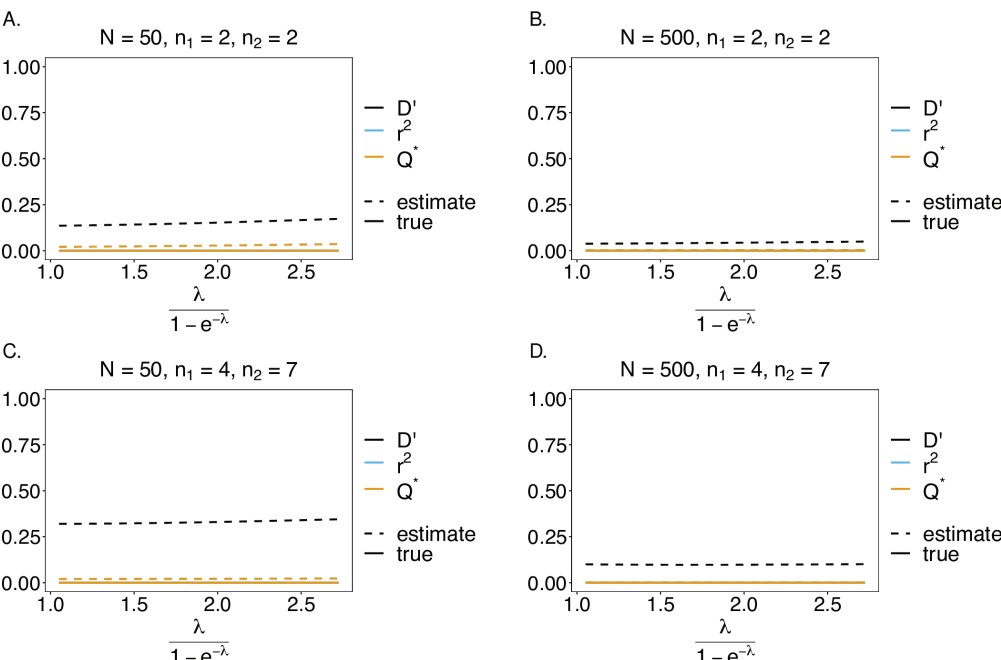

**Fig 10. Linkage disequilibrium (LD) – the balanced case**: Shown are estimates of LD plotted against the mean MOI. The balanced distributions of Table 1 for the genetic architectures $n_1 = n_2 = 2$ (A, B) and $n_1 = 4$, $n_2 = 7$ (C, D) are assumed. The estimates presented are for small and large sample sizes, i.e., $N = 50$ (A, C) and $N = 500$ (B, D), respectively. Colors correspond to different LD measures. The solid and dashed lines show the true and estimated LD values.

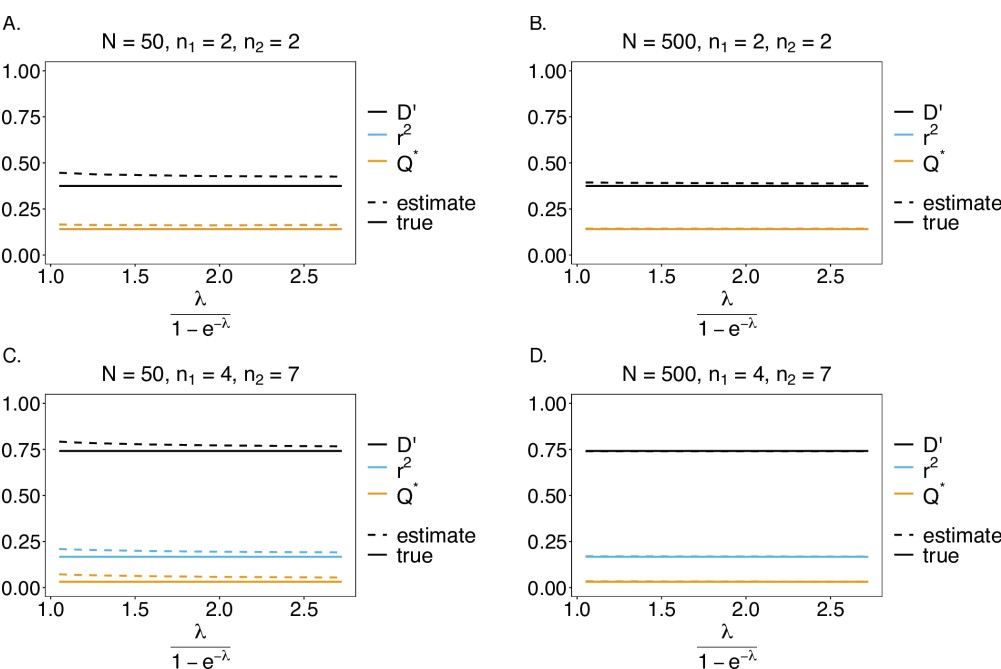

**Fig 11. Linkage disequilibrium (LD) – the unbalanced case**: Shown are estimates of LD plotted against the mean MOI. We assume the unbalanced haplotype distributions of Table 1 for the genetic architectures $n_1 = n_2 = 2$ (A, B) and $n_1 = 4$, $n_2 = 7$ (C, D). The estimates presented are for small and large sample sizes, i.e., $N = 50$ and $N = 500$, respectively. Colors correspond to different LD measures, while the solid line shows the true value of LD and the dashed line shows the estimates.

## Discussion

Disease surveillance, enhanced by genetic/molecular techniques, became increasingly feasible and popular to monitor infectious diseases [38] such as malaria [16]. Due to considerable efforts and resources devoted to malaria control, the disease burden was reduced substantially in the last two decades [8], with some endemic areas shifting their goals from disease control to elimination. However, there is an upward trend in worldwide malaria incidence and mortality since 2018 and the disease remains highly prevalent in many countries, particularly in Sub-Saharan Africa [8]. Areas with high malaria endemicity harbor substantial genetic variation, e.g., caused by ectopic recombination in the var gene families [39], which needs to be constantly monitored. Importantly, genetic variation in the pathogen, e.g., msp1 and msp2 in *P. falciparum* can affect disease severity [40]. Furthermore, when aiming for eradication, transmission intensities and routes of transmission have to be monitored closely. Additionally, successful malaria control and eradication attempts are challenged by the emergence and spread of antimalarial-drug resistance and *Pf*HRP2/3 gene deletions [8,41,42].

In malaria, molecular surveillance became widely used to monitor genetic diversity [16]. In areas of high transmission, the genetic diversity of pathogen antigens is relevant for the clinical presentation of the disease. Moreover, changes in genetic diversity can reflect the effectiveness of control interventions [43], with decreasing diversity being indicative of successful control measures, and vice versa. Regarding the latter, the increase in genetic diversity in Venezuela between 2004 and 2017, after the economic situation worsened and affected malaria control programs, was obvious and particularly measurable by MOI [44]. In areas of low transmission, molecular surveillance can be informative of routes of transmission, as was

described in Colombia [45]. This is particularly important in the context of elimination, to distinguish whether local disease outbreaks are caused by migrational events reflecting deficits in measures to contain outbreaks locally, or from relapses or recrudescence being indicative of deficits in diagnostics or treatments. Diagnostic failures can be the result of the widespread prevalence of *Pf*HRP2/3 deletions, while recrudescence can be a result of widespread drug resistance. Molecular methods are not just adequate to estimate the frequency/prevalence of resistance, e.g., [19,46], but also to explain and reconstruct the history of drug-resistance evolution [20]. Importantly, the population genetics of *Plasmodium* differs from standard population genetics, due to the organism's specific transmission cycle [47–49]. More precisely, the processes of selection and recombination cannot be decoupled [50]. This can even result in genome-wide reductions of genetic variation caused by selection in low transmission areas, i.e., in areas with low MOI [15,50]. Similarly, patterns of linkage disequilibrium (LD) indicative for selection, have to be carefully interpreted in the context of the distribution of MOI. LD is expected to be higher in areas with low malaria endemicity [51], because the effective recombination rates are reduced by low MOI.

To adequately estimate pairwise LD in malaria and similar diseases, we introduced a statistical model to estimate the frequency of pathogenic variants alongside MOI from a pair of multi-allelic marker loci (with $n_1$ and $n_2$ alleles segregating at the first and second marker, respectively). This genetic architecture can be interpreted differently. A locus can be an STR (microsatellite) marker (as in the empirical example provided), a SNP which is not necessarily bi-allelic, or a micro haplotype, i.e., a set of DNA sequences or a panel of SNPs in short non-recombining regions, for which phased molecular data can be obtained. The latter is increasingly feasible by third-generation sequencing methods [52]. In this context, MOI follows the definition of [17], i.e., as the number of super-infections during one disease episode. This neglects the possibility of co-transmission of several pathogenic variants at one infectious event (co-infection). The type of molecular data to which the proposed method is applicable does not provide enough resolution to adequately address co-infections. For more discussion and justification of this assumption, see [17].

The proposed method provides maximum-likelihood estimates (MLE) of (two-locus) haplotype frequencies and the distribution of MOI, assuming an underlying Poisson distribution. The latter assumption conveniently reduces MOI to a single parameter. The frequency estimates can be used as plugin estimates to derive LD measures, which have to be interpreted in the context of MOI. The statistical model is too complex to allow a closed-form solution to derive the MLE. Therefore, the expectation-maximization (EM) algorithm was employed, which provides a numerically stable method to derive the MLE. Because the parameter space is constrained, i.e. the frequencies are elements of an $(n_1 n_2 - 1)$-dimensional simplex and the MOI parameter is positive, Newton-Raphson methods harbor the problem that the parameter updates might fall out of the admissible parameter space. Specifically, such methods are sensitive to initial parameter choices. The EM-algorithm does not share this problem, but convergence might be slow close to the maximum. The numerical investigations revealed that computational time is sufficiently fast (cf. [53] Chapter 2).

In practice, the method can be readily applied for malaria disease surveillance as a replacement for heuristic methods. Multi-allelic neutral markers are ideal for estimating MOI. Because this quantity scales with transmission intensity, by temporal comparison, it can be adapted as a measure to evaluate the impact of disease control efforts. Specifically, by annual active or passive surveillance, MOI can be estimated each year. If control efforts effectively reduce transmission, MOI should decrease over time. The convenience is that no elaborate sample design is required, as parasites randomly infect humans. The choice of neutral markers is important in this regard, as they should not influence disease symptoms. MOI is

a more adequate measure in this regard than epidemiological records. Namely, as transmission reduces, also the levels of host-acquired immunity in a particular endemic area decrease, leading to more symptomatic infections. Hence, scaling up control efforts might paradoxically lead to an increase in the number of diagnosed cases. An additional factor is that screening efforts are often upscaled as part of control efforts. This is witnessed from the latest World Malaria Report [54]. Namely, while the number of estimated cases substantially decreased in the past 25 years, this is not true for the number of diagnosed cases. MOI is not affected by these factors. Likewise, MOI can be used to evaluate control efforts across endemic areas on a spatial scale. Patterns of LD and genetic diversity across time and/or endemic regions can be used to identify migrational events and thereby identify critical nodes of malaria transmission. An advantage of the proposed method is that it can estimate pairwise LD or genetic diversity without deflating sample size as in heuristic methods, thereby limiting uncertainty in the estimates. This is particularly useful when deriving LD maps from data with many ambiguous observations due to MOI.

Concerning the genetic architecture, the numbers of segregating alleles $n_1$ and $n_2$ are arbitrary. However, the method might suffer from the curse of dimensionality if the number of alleles is too large. For instance, for typical sample sizes $N$, up to 30 alleles can be commonly observed at highly variable STR markers. As an example, $n_1 = n_2 = 30$ leads to 900 possible haplotypes (most of which will not even be realized in the population), such that the number of model parameters by far exceeds realistic sample sizes. Notably, in practice, the numerical implementation of the method is not affected by the curse of dimensionality, because typically only a few alleles are observed in a sample at each marker. However, for pathologic examples, implementations of the EM algorithm might exceed the RAM of standard desktop computers. The computational limitations of the methods depend crucially on the sets $\mathcal{A}_{\boldsymbol{x}}$ to be constructed. This depends on the genetic architecture and the underlying dataset. For instance, consider a genetic architecture with $n_1 = n_2 = 12$ alleles at each marker, yielding 144 possible haplotypes. In the worst case, a set $\mathcal{A}_{\boldsymbol{x}}$ contains up to 16 769 025 elements, which is still feasible. In contrast, for a genetic architecture with $n_1 = 72$ and $n_2 = 2$ alleles, which also yields 144 possible haplotypes, the sets $\mathcal{A}_{\boldsymbol{x}}$ contain up to more than $1.4 \times 10^{22}$ elements, which exceeds normal RAM by orders of magnitude. However, in both genetic architectures, the method is feasible in datasets that contain less than 10 alleles at each marker in every observation. Considering the running time of the method, for a genetic architecture with $n_1 = 5$, and $n_2 = 2$ alleles, the running time for 10 000 datasets is around 1 262 seconds, or roughly 20 min. The running time depends also on the underlying MOI parameter. In general, higher MOI creates datasets with more evidence of super-infections. For such datasets, the running time is longer. In any case, while the method has its computational limitations, it is suitable for datasets that occur in practice.

When creating pairwise LD maps (e.g., Figs 4, 5) between $L$ markers, the method needs to be applied $\frac{L(L-1)}{2}$ times, thereby yielding $\frac{L(L-1)}{2}$ estimates for the MOI parameter and the allele frequencies of each marker can be obtained by marginalization from $L-1$ of the two-marker estimates. While it is undesirable not to obtain a single estimate for the MOI parameter, it is justified by three reasons. First, if only pairwise LD is of interest, the MOI estimates are of secondary importance. Second, one could average the $\frac{L(L-1)}{2}$ estimates, and use this in a second step as a common plugin estimate for a new estimation of all $\frac{L(L-1)}{2}$ two-marker frequencies, thereby heuristically avoiding the different MOI estimates. Third, a full-haplotype-based method, which derives the frequency distribution for haplotypes characterized by all available markers (and then all two-marker distributions by marginalization) can be undesirable due to the curse of dimensionality, or because of missing data. For the former, assume $L = 10$

markers with 5 alleles segregating at each marker in a sample of size $N$ = 200. This results in 9 765 625 possible haplotypes by far exceeding sample size, while there are only 25 possible two-marker haplotypes for each parameter combination, being much smaller than sample size. For the latter, in practice molecular assays sometimes fail, resulting in missing values. In a full-haplotype-based method, only samples can be used, for which molecular assays produce data at all markers. For each 2-marker comparison, more samples can be retained from the data than for a full-haplotype-based method.

The quality of the proposed method was investigated by numerical simulations and showed overall desirable properties. Particularly, bias is low, and the simulation suggests asymptotic unbiasedness. This was explicitly proven for the corresponding method using a single molecular marker [27] and is also in agreement with the corresponding method using a genetic architecture of multiple bi-allelic (SNP) markers [19]. Note, in the case of a single molecular marker, the model falls in the class of exponential families [55], by which the asymptotic properties of the estimator follow. With two or more markers, there is no obvious way to rewrite the model as an exponential family, and it is questionable whether it is possible. Therefore, the asymptotic properties do not immediately follow. However, the results from the single-marker case and the numerical investigations performed here, support that the estimator has desirable small- and large-sample properties.

The quality of LD estimates was assessed through the standard measures, namely, $D'$, $r^2$, $Q^*$, and ALD. In general, multi-allelic LD measures are more difficult to interpret than biallelic LD measures [22]. All these measures harbor limitations. Nevertheless, these are standard in evolutionary-genetic analyses and the performance of the proposed method to obtain plugin estimates for LD measures is sufficiently accurate. However, caution is required when choosing an appropriate LD measure.

As an example, we applied the method to an empirical dataset from Cameroon consisting of molecular information at markers flanking the *Pfdhfr* and *Pfdhps* loci in *P. falciparum* malaria, associated with drug resistance. The application confirmed that the method properly captures the intuitively expected behavior, giving further evidence for the appropriateness of the method.

The method is implemented in an easy-to-use R script available as supporting information, also on GitHub at https://github.com/Maths-against-Malaria/MultiAllelicBiLociModel (this version will be further maintained) and on Zenodo at https://doi.org/10.5281/zenodo.8289710. For a description and examples of how to use the R script, see S1 User Manual and https://github.com/Maths-against-Malaria/MultiAllelicBiLociModel.

## Supporting information

**S1 Fig. Linkage disequilibrium at SNP markers**: Shown are estimates of pairwise LD for six SNP marker loci associated with SP drug-resistance, i.e., three codons (51, 59, 108) at *Pfdhfr* and three codons (436, 437, 613) at *Pfdhps*. Panels (A, B) show $r^2$ values, whereas (C, D) show $Q^*$ values. Furthermore, panels (A, C) correspond to the years 2001/2002, while panels (B, D) correspond to the years 2004/2005. The codons on *Pfdhfr* and *Pfdhps* are highlighted on the maps by horizontal black lines, with the name of the gene and codons are specified on top and below the line, respectively. The thick black lines in panels (A, B) group pairwise LD within each gene. The numbers indicate LD values.
(TIF)

**S2 Fig. Linkage disequilibrium at microsatellite markers**: As in S1 Fig but for a set of genes at two neutral chromosomes, i.e., chromosomes 2 and 3, genes around *Pfdhfr* on chromosome 4, and genes around *Pfdhps* on chromosome 8.
(TIF)

**S1 Mathematical appendix. Mathematical background.**
(XLS)

**S1 User Manual. Description of the usage of the method's implementation.**
(XLS)

**S1 R Script and example datasets. Zip file containing an R script containing the implementation of the method and a few example datasets.**
(XLS)

## Acknowledgments

The molecular dataset used for data applications was provided by Dr. Andrea M. McCollum and Dr. Venkatachalam Udhayakumar, which is gratefully acknowledged. We also want to thank two anonymous reviewers for the encouraging and constructive comments on the manuscript.

## Author contributions

**Conceptualization:** Henri Christian Junior Tsoungui Obama, Kristan Alexander Schneider.

**Data curation:** Henri Christian Junior Tsoungui Obama.

**Formal analysis:** Henri Christian Junior Tsoungui Obama, Kristan Alexander Schneider.

**Funding acquisition:** Kristan Alexander Schneider.

**Investigation:** Henri Christian Junior Tsoungui Obama, Kristan Alexander Schneider.

**Methodology:** Henri Christian Junior Tsoungui Obama, Kristan Alexander Schneider.

**Project administration:** Kristan Alexander Schneider.

**Resources:** Kristan Alexander Schneider.

**Software:** Henri Christian Junior Tsoungui Obama, Kristan Alexander Schneider.

**Supervision:** Kristan Alexander Schneider.

**Validation:** Henri Christian Junior Tsoungui Obama, Kristan Alexander Schneider.

**Visualization:** Henri Christian Junior Tsoungui Obama.

**Writing – original draft:** Henri Christian Junior Tsoungui Obama, Kristan Alexander Schneider.

**Writing – review & editing:** Henri Christian Junior Tsoungui Obama, Kristan Alexander Schneider.

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
