## [Decision Letter · Decision Letter 0]

14 Jan 2025

PONE-D-23-29177Estimating multiplicity of infection, haplotype frequencies, and linkage disequilibria from multi-allelic markers for molecular disease surveillancePLOS ONE

Dear Dr. Tsoungui Obama,

Thank you for submitting your manuscript to PLOS ONE. After careful consideration, we feel that it has merit but does not fully meet PLOS ONE’s publication criteria as it currently stands. Therefore, we invite you to submit a revised version of the manuscript that addresses the points raised during the review process.

We look forward to receiving your revised manuscript.

Kind regards,

Segun Isaac OYEDEJI, Ph.D

Academic Editor

PLOS ONE

Journal Requirements:

2. Please include a copy of Table 3 which you refer to in your text on page 15.

3. We are unable to open your Supporting Information file [STRModel.R, STRmle.R]. Please kindly revise as necessary and re-upload.

**Additional Editor Comments:**

In addition, please attend to the areas in need of minor corrections as outlined below.

Thank you.

INTRODUCTION

Page 3: Line 85

Please correct the clause "...alleles at makers of concern" to "...alleles at markers of concern".

Page 3: Lines 91/92

Please correct the clause "...considering one polymorphic maker locus" to "...considering one polymorphic marker locus".

METHODS

Page 3: Line 131

Please correct the clause "...two multialelic" to "...two multiallelic"

Reviewers' comments:

Reviewer's Responses to Questions

**Comments to the Author**

1. Is the manuscript technically sound, and do the data support the conclusions?

Reviewer #1: Yes

Reviewer #2: Yes

2. Has the statistical analysis been performed appropriately and rigorously? 

Reviewer #1: Yes

Reviewer #2: Yes

3. Have the authors made all data underlying the findings in their manuscript fully available?

Reviewer #1: Yes

Reviewer #2: Yes

4. Is the manuscript presented in an intelligible fashion and written in standard English?

Reviewer #1: Yes

Reviewer #2: Yes

5. Review Comments to the Author

Reviewer #1: The manuscript in its present form is technically sound as it provides the mathematical models. It is well written in clear and understandable language. The analysis is also adequate. The manuscript can therefore be published in its present form.

Reviewer #2: The manuscript presents a statistical framework for estimating haplotype frequencies and multiplicity of infection (MOI) from multi-allelic molecular markers, leveraging the maximum-likelihood method and the expectation-maximization (EM) algorithm. The study is well-motivated, addressing critical challenges in molecular disease surveillance, particularly in diseases like malaria. It demonstrates methodological rigor and is complemented by numerical simulations and a real-world application to sulfadoxine-pyrimethamine resistance in Plasmodium falciparum from Cameroon. The manuscript offers an innovative solution to a complex problem and makes a valuable contribution to the field.

See also the attached document.

6. PLOS authors have the option to publish the peer review history of their article (what does this mean?). If published, this will include your full peer review and any attached files.

Reviewer #1: No

Reviewer #2: No

---

## [Author Response · Author response to Decision Letter 1]

18 Feb 2025

Editor Comments:

INTRODUCTION

Page 3: Line 85

Please correct the clause "...alleles at makers of concern" to "...alleles at markers of concern”.

Response: Fixed.

Page 3: Lines 91/92

Please correct the clause "...considering one polymorphic maker locus" to "...considering one polymorphic marker locus”.

Response: Fixed.

METHODS

Page 3: Line 131

Please correct the clause "...two multialelic" to "...two multiallelic”

Response: Fixed.

Reviewer #1:

No comments

Response: We are grateful for the appreciation of our work.

Reviewer #2:

Overall Summary: The manuscript presents a statistical framework for estimating haplotype frequencies and multiplicity of infection (MOI) from multi-allelic molecular markers, leveraging the maximum-likelihood method and the expectation-maximization (EM) algorithm. The study is well-motivated, addressing critical challenges in molecular disease surveillance, particularly in diseases like malaria. It demonstrates methodological rigor and is complemented by numerical simulations and a real-world application to sulfadoxine-pyrimethamine resistance in Plasmodium falciparum from Cameroon. The manuscript offers an innovative solution to a complex problem and makes a valuable contribution to the field.

Response: We appreciate the positive feedback! We tried to incorporate all suggestions. In the following, our responses are just for the suggested changes. 

Technical Feedback:

Methodological Soundness:

The use of the EM algorithm to estimate haplotype frequencies and MOI is well-justified, given the absence of a closed-form solution for maximum-likelihood estimates. The mathematical derivations are comprehensive and technically accurate.

The adaptation of the method to account for multi-allelic loci and ambiguous observations due to MOI is a strong innovation. This addresses a key gap in traditional population genetics tools, which often fail to handle polyploid-like behavior in malaria infections effectively.

Response: We appreciate this assessment.

Simulation Study:

The choice of parameters (e.g., allele numbers, sample sizes, MOI distributions) is appropriate and relevant for real-world malaria data. The extensive simulations across balanced and unbalanced haplotype frequency distributions provide robust evidence of the method’s accuracy and precision.

However, the manuscript could benefit from additional discussion on computational efficiency. While the EM algorithm is known for stability, its convergence in high-dimensional parameter spaces can be slow. Quantitative metrics on runtime performance would enhance the utility of the work.

Response: This is a great suggestion! It is important to note that the running depends on several factors. 1. There is a crucial dependency on the underlying dataset. Namely, in several steps the sets A_x (eq. 3b) have to be constructed. The cardinality of these sets vary substantially. If x corresponds to a single infection, A_x has just one element. However, if the observation x all alleles at both loci, the cardinality is (2^(n_1)-1)*(2^(n_2)-1). So for n1=n2=20 alleles (which can occur in practice for STRs) this would be more than 10^12 elements, which substantially increases running time. 2. In the iteration of the EM algorithm, we observed that the convergence of the Poisson parameter is low from the left, but much faster from the right. Hence, the initial condition needs to be sufficiently large. The number of iterations can vary from a handful to millions. We built in a heuristic that tries different initial conditions if convergence is too slow. While this typically does not matter when deriving the estimates for a single dataset, it made a substantial difference for the simulations, since the estimates are derived for thousands of datasets. This contributed substantially to increase the speed of the simulations. We added some discussion on the running time of the algorithm and its limitations with some examples, but tried to keep it brief. 

Real-World Application:

The application to sulfadoxine-pyrimethamine resistance in P. falciparum is well-aligned with the study’s objectives and demonstrates the method’s relevance. The LD maps effectively capture evolutionary dynamics and drug pressure effects.

The discussion around high LD in regions flanking Pfdhfr and Pfdhps is compelling. However, the manuscript could explore potential confounding factors in the observed LD patterns, such as sampling bias or population substructure.

Response: See next response. 

Linkage Disequilibrium Measures:

The use of multiple LD measures (D′, r², Q∗, and asymmetric conditional LD) is a strength, as it offers diverse perspectives on genetic associations. The manuscript could further elaborate on the comparative advantages and limitations of these measures, especially for multi-allelic loci.

Response: This is a good suggestion. We added which artifacts of D’ occur in the present data set, and explained that in detail. 

Clarity:

Abstract and Introduction:

The abstract succinctly summarizes the study’s objectives, methods, and key findings. However, it could briefly mention the practical implications of the findings, particularly their relevance for disease control policies.

The introduction effectively sets the stage by highlighting the challenges of MOI and LD estimation in malaria. The transition from problem statement to proposed solution is clear and logical.

Response: We appreciate the point. In fact, the conclusions part in the abstract was a bit short. We hence added some practical implications for molecular disease surveillance in the abstract. 

Methodology:

The mathematical details in the methods section are thorough but may be dense for readers unfamiliar with advanced statistical modeling. Consider adding a conceptual overview or visual aids (e.g., flowcharts) to complement the equations.

Response: We agree with the assessment. Our approach to the presentation was to facilitate readability by a broad audience. As a consequence, we moved the mathematically more involved part into the appendix, which limits the logical flow to a certain extent. We hence added a new figure which gives a conceptual overview of the manuscript and indicates which background knowledge is required and which sections can be skipped if not interested in technical details. 

Definitions of key terms (e.g., MOI, haplotype frequencies) are clear, but the distinction between “super-infections” and “co-infections” could be expanded for clarity.

Response: this is an important point, which deserves careful consideration. Particularly, verbal explanations are not always ideal, as they might make the difference between “super-infections” and “co-infections” clear. Hence, we included a new figure, which illustrates the difference, between “super-infections”, “co-infections”, and a mixture of both. 

Results and Figures:

Figures are well-designed and enhance the manuscript’s readability. The LD maps (Figures 2 and 3) are particularly informative. Adding annotations to highlight key observations could further improve their interpretability.

Response: we tried to add annotations to the LD figure, but could not come up with anything aesthetically pleasing. We hence tried to highlight the context by verbal descriptions. 

The text accompanying the results is concise, but some sections (e.g., bias and variance analysis) would benefit from additional context to aid readers in interpreting the findings.

Response: This is a good suggestion. We added some additional context. 

Discussion and Conclusion:

The discussion is well-structured, linking the findings to broader implications for molecular disease surveillance. It appropriately acknowledges limitations, such as the impact of small sample sizes and unbalanced haplotype frequencies.

The conclusion effectively reiterates the study’s contributions but could offer more specific recommendations for future research or practical applications.

Response: This suggested we added more recommendations and discussion of how the method can be applied in practice. 

Novelty:

The manuscript addresses a significant gap in the estimation of MOI and haplotype frequencies from multi-allelic data, which is critical for molecular disease surveillance.

The integration of multi-allelic LD measures and the application to real-world malaria data highlight the framework’s versatility and practical value.

The implementation of the method as an open-source R script enhances its accessibility and potential for adoption by the research community.

Suggestions for Improvement:

Expand on Practical Applications: Provide examples of how the proposed method can inform public health strategies, such as monitoring drug resistance or optimizing intervention programs.

Response: This was addressed, see above.

Enhance Accessibility: Include a flowchart summarizing the methodological steps and a brief guide for non-specialists to understand the key concepts.

Response: This was incorporated, see above.

Address Computational Aspects: Discuss runtime performance and scalability for larger datasets, as this is a critical consideration for field applications.

Response: This was addressed, see above.

Supplementary Materials: While the manuscript mentions a user manual, including a brief tutorial in the main text on using the R script would make the method more accessible.

Response: This was addressed, see above.

---

## [Editor Report · Decision Letter 1]

12 Mar 2025

Estimating multiplicity of infection, haplotype frequencies, and linkage disequilibria from multi-allelic markers for molecular disease surveillance

PONE-D-23-29177R1

Dear Dr. Tsoungui Obama,

We’re pleased to inform you that your manuscript has been judged scientifically suitable for publication and will be formally accepted for publication once it meets all outstanding technical requirements.

Kind regards,

Segun Isaac OYEDEJI, Ph.D

Academic Editor

PLOS ONE

---

## [Editor Report · Acceptance letter]

PONE-D-23-29177R1

PLOS ONE

Dear Dr. Tsoungui Obama,

I'm pleased to inform you that your manuscript has been deemed suitable for publication in PLOS ONE. Congratulations! Your manuscript is now being handed over to our production team.

Kind regards,

on behalf of

Professor Segun Isaac OYEDEJI

Academic Editor

PLOS ONE